# SSTAG: Structure-Aware Self-Supervised Learning Method for Text-Attributed Graphs

**Ruyue Liu**
Institute of Information Engineering, CAS
School of Cyberspace Security, UCAS
liuruyue@iie.ac.cn

**Rong Yin**[*]
Institute of Information Engineering, CAS
School of Cyberspace Security, UCAS
yinrong@iie.ac.cn

**Xiangzhen Bo**
Wuhan University of Technology
353145@whut.edu.cn

**Xiaoshuai Hao**
Xiaomi EV
haoxiaoshuai@xiaomi.com

**Yong Liu**
Renmin University of China
liuyonggsai@ruc.edu.cn

**Jinwen Zhong**
Institute of Information Engineering, CAS
zhongjinwen@iie.ac.cn

**Can Ma**
Institute of Information Engineering, CAS
macan@iie.ac.cn

**Weiping Wang**
Institute of Information Engineering, CAS
wangweiping@iie.ac.cn

## Abstract

Large-scale pre-trained models have revolutionized Natural Language Processing (NLP) and Computer Vision (CV), showcasing remarkable cross-domain generalization abilities. However, in graph learning, models are typically trained on individual graph datasets, limiting their capacity to transfer knowledge across different graphs and tasks. This approach also heavily relies on large volumes of annotated data, which presents a significant challenge in resource-constrained settings. Unlike NLP and CV, graph-structured data presents unique challenges due to its inherent heterogeneity, including domain-specific feature spaces and structural diversity across various applications. To address these challenges, we propose a novel structure-aware self-supervised learning method for Text-Attributed Graphs (**SSTAG**). By leveraging text as a unified representation medium for graph learning, SSTAG bridges the gap between the semantic reasoning of Large Language Models (LLMs) and the structural modeling capabilities of Graph Neural Networks (GNNs). Our approach introduces a dual knowledge distillation framework that co-distills both LLMs and GNNs into structure-aware multilayer perceptrons (MLPs), enhancing the scalability of large-scale TAGs. Additionally, we introduce an in-memory mechanism that stores typical graph representations, aligning them with memory anchors in an in-memory repository to integrate invariant knowledge, thereby improving the model's generalization ability. Extensive experiments demonstrate that SSTAG outperforms state-of-the-art models on cross-domain transfer learning tasks, achieves exceptional scalability, and reduces inference costs while maintaining competitive performance.

---

[*]Corresponding author.

# 1 Introduction

In recent years, large-scale pre-trained models have achieved revolutionary breakthroughs in natural language processing (NLP) [1] and computer vision (CV) [2], demonstrating remarkable cross-domain generalization capabilities [3]. However, the prevailing paradigm in graph learning remains confined to training dedicated models for individual graph datasets [4, 5]. This single-graph modeling approach suffers from two major limitations: (1) models are typically restricted to single or narrowly defined tasks, lacking the ability to transfer knowledge across different graphs; and (2) model performance heavily depends on the scale of annotated data, yet acquiring high-quality labels is often costly and time-consuming, creating a significant bottleneck in low-resource scenarios.

The success of foundation models in language and vision stems from their domain invariance, supported by unified lexical or pixel spaces. In contrast, developing graph foundation models is challenging due to the intrinsic heterogeneity of graph-structured data. Graphs exhibit domain-specific node and edge types, with diverse feature and label spaces, making cross-domain alignment difficult. Moreover, structural diversity, such as acyclic citation networks versus multi-relational cyclic knowledge graphs, further complicates knowledge transfer across domains. To address these challenges, we leverage text as a unified representation medium for graph learning. Many real-world graphs are inherently text-attributed, where raw textual features provide a domain-agnostic semantic space. Large language models (LLMs) excel in semantic understanding and reasoning [6, 7, 8], but struggle with topological reasoning [9, 10], where graph neural networks (GNNs) excel. Conversely, GNNs lack the open-world knowledge embedded in LLMs, motivating a unified framework that bridges their complementary strengths.

To bridge this gap, we propose a novel **S**tructure-aware **S**elf-supervised learning method for **T**ext-**A**ttributed **G**raphs, called **SSTAG**. Specifically, to learn transferable invariants across graphs and tasks, we design a generic template that unifies various tasks by contextualizing the nodes, edges, and graphs for which we make predictions. For node or edge-level tasks on large-scale graphs, we employ the Personalized PageRank (PPR) algorithm to sample subgraphs, which mitigates the differences in graph structure across domains and enhances the scalability of the model. Additionally, we introduce a new pre-training objective of co-distilling language models (LMs) and graph neural networks (GNNs) into structure-aware multilayer perceptrons (MLPs), specifically tailored for self-supervised learning on large-scale task-attribute graphs (TAGs). This approach offers a dual advantage: (1) Through multimodal distillation, the MLP absorbs both the structural modeling capabilities of GNNs and the semantic reasoning abilities of LLMs. (2) The lightweight MLP circumvents the high computational overhead of LLMs, making it more suitable for practical deployment. This two-stage knowledge transfer paradigm not only overcomes the domain limitations of single graph models but also mitigates the structural processing limitations inherent in pure LLM approaches.

To summarize, our main contributions are as follows:

(1) We propose a general-purpose graph learning framework that unifies node-, edge-, and graph-level prediction tasks within a single architecture. The unified design enables flexible adaptation and effective knowledge transfer across heterogeneous graph domains and diverse downstream tasks, overcoming the limitations of task-specific and domain-isolated models.

(2) We design a novel self-supervised pretraining objective that distills complementary knowledge from large language models (LLMs) and graph neural networks (GNNs) into a structure-aware multi-layer perceptron (MLP), combining semantic reasoning with structural understanding while ensuring efficient inference.

(3) Extensive experiments conducted on multiple benchmark datasets demonstrate the superiority of our proposed SSTAG framework: (a) it outperforms state-of-the-art baselines on cross-domain transfer learning tasks; (b) it exhibits remarkable scalability on large-scale graphs compared to existing GNN and LLM-based methods;(c) it significantly reduces inference cost while maintaining competitive performance.

# 2 Related Work

**Representation Learning on TAGs** Research on Text-Attributed Graphs (TAGs) lies at the intersection of graph machine learning and natural language processing. Early approaches focused on

shallow text-based enhancements for graph embeddings [11, 12], where textual features are treated as auxiliary node attributes within traditional graph algorithms. While computationally efficient, these methods fail to capture the deep semantic interplay between textual content and graph structures. Another class of graph learning models based on TAGs are LLMs-only approaches, such as LLaGA [13] and GraphGPT [14]. These methods leverage instruction tuning to map graph-structured data into the embedding space of large language models. The emergence of graph neural networks [15] revolutionizes TAGs processing by enabling end-to-end representation learning. For example, TAPE [16] leverages large language models to generate explanatory node descriptions, which are then used as enriched features for training GNNs. Graph-LLM [17] converts graph structures into textual sequences for downstream prediction via LLMs. Das et al. [18] explore the integration of graph data with LLMs, along with the influence of multi-modal representations. CaR [19] extracts textual captions from molecular SMILES strings using LLMs and feeds them into another language model for fine-tuning. However, they primarily rely on supervised training, which limits their applicability in low-resource or unlabeled scenarios.

**Self-Supervised Learning on Graphs**   Self-supervised learning has emerged as a compelling paradigm for learning representations from graph-structured data without the need for explicit labeling. Existing work in this area can be broadly classified into two main categories: contrastive learning methods and generative methods. Contrastive learning methods aim to learn graph representations by maximizing the similarity between positive pairs while minimizing the similarity between negative pairs. GraphCL [20] has significantly advanced contrastive learning techniques by introducing various graph data augmentation strategies. These methods typically rely on effective strategies for pairing positive and negative samples, along with robust GNN architectures to extract meaningful graph features. More recently, methods like GPA [21] have introduced personalized graph enhancement strategies to further improve the quality of learned representations. Generative methods, on the other hand, focus on learning graph representations by predicting the missing or unobserved parts of the graph. For instance, GraphMAE [22] employs GNN-based encoders and decoders to reconstruct masked node features, while S2GAE [23] uses a similar approach to mask edges within the graph and predict the missing links. However, these methods remain confined to single-graph settings and face significant challenges in achieving cross-domain generalization.

## 3   Preliminaries

**Text-Attributed Graphs**   Given a text-attributed graph $\mathcal{G} = \{\mathcal{V}, \mathcal{E}, \mathcal{T}_\mathcal{V}, \mathcal{T}_\mathcal{E}, \boldsymbol{A}\}$ with $N$ nodes, where $\mathcal{V}$ represents the set of nodes and $\mathcal{E}$ represents the set of edges. For each node $v \in \mathcal{V}$, there is an associated text $t_v \in \mathcal{T}_\mathcal{V}$ that represents the node-level textual information. For each edge $e_{vu} \in \mathcal{E}$ connecting nodes $v$ and $u$, there is an associated text $t_{e_{vu}} \in \mathcal{T}_\mathcal{E}$ that represents the edge-level textual information. The adjacency matrix is denoted as $\boldsymbol{A} \in \mathbb{R}^{N \times N}$. In this work, we focus on self-supervised learning on text-attributed graphs (TAGs). Specifically, the goal is to pre-train a mapping function $f_\theta : \mathcal{T}_\mathcal{V} \times \boldsymbol{A} \to \mathbb{R}^d$ or $\mathcal{T}_\mathcal{E} \times \boldsymbol{A} \to \mathbb{R}^d$ such that the semantic information in $\mathcal{T}_\mathcal{V}$ or $\mathcal{T}_\mathcal{E}$ and the topological information in $\boldsymbol{A}$ can be efficiently captured in a $d$-dimensional space in a self-supervised manner.

**Graph Neural Networks**   For graph-structured data, Graph Neural Networks (GNNs) are commonly used to instantiate $f_g$. Specifically, the objective of GNNs is to update node representations by aggregating messages from their neighbors, as expressed by the following equation: $h_v^{(k)} = \text{COM}\left(h_v^{(k-1)}, \text{AGG}\left(\{h_u^{(k-1)} : u \in \mathcal{N}(v)\}\right)\right)$, where $h_v^{(k)}$ represents the representation of node $v$ at the $k$-th layer, and $\mathcal{N}(v) = \{u \mid \boldsymbol{A}_{v,u} = 1\}$ is the set of one-hop neighbors of node $v$. In particular, we have $h_v^{(0)} = x_v$, where $x_v = \text{Emb}(t_v) \in \mathbb{R}^F$ is a $F$-dimensional feature vector extracted from the textual attributes $t_v$ of nodes, and $\text{Emb}(\cdot)$ denotes the embedding function. The AGG function is used to aggregate features from the neighbors, while the COM function combines the aggregated neighbor information with the own node embedding from the previous layer.

## 4   Proposed Method

In this section, we propose SSTAG, a novel framework designed to learn robust and informative graph representations by integrating structural and textual signals in a self-supervised manner. As shown in Figure 1, the proposed method comprises three key components: the Unified Graph Task (UGT)

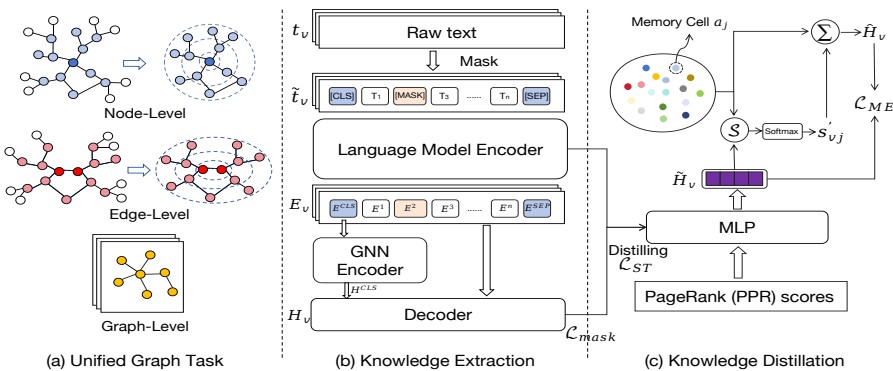

Figure 1: Overview of the proposed SSTAG framework.

module, the Knowledge Extraction from LLM (KEL) module, and the Knowledge Distillation(KD) module. Given a text-attributed graph, SSTAG first constructs a generic and task-agnostic self-supervised objective via the UGT module, which encodes both node structure and attribute semantics. Subsequently, the KEL leverages a LLM to capture high-level semantic representations from the node-associated textual attributes. These representations are aligned with graph-based representations obtained from a GNN.To effectively bridge the modality gap between language and graph features, we introduce a Knowledge Distillation module that transfers the complementary knowledge from both the LLM and the GNN into a lightweight MLP, enabling efficient downstream adaptation. Finally, the pre-trained SSTAG model can be fine-tuned for various downstream tasks at different granularity levels, such as node classification, link prediction, and graph classification.

## 4.1 Unified Graph Task

Graphs from different domains often exhibit diverse structural patterns and serve distinct application scenarios and task objectives. To address this heterogeneity, graph learning tasks are typically categorized into three levels based on structural granularity: node-level tasks, edge-level tasks, and graph-level tasks. Recent studies suggest that subgraph-based representations offer notable advantages. On one hand, they enhance the expressive capacity of models by incorporating richer local structures [24, 25]; on the other, they enable standardized task formulation across different levels [26]. Motivated by this, we adopt a unified representation format that leverages target nodes along with their corresponding context subgraphs.

**Node-Level Tasks** We design a subgraph sampling strategy that integrates the Personalized PageRank (PPR) algorithm [27]. For a given node $v$, its importance score $\pi_v$ is computed as follows:

$$\pi_v = \alpha(\boldsymbol{I} - (1-\alpha)\tilde{\boldsymbol{A}})^{-1}\boldsymbol{e}_v, \tag{1}$$

where $\boldsymbol{I}$ is the unit matrix, $\tilde{\boldsymbol{A}}$ denotes the normalized adjacency matrix, $\alpha$ is the teleport factor, and $\boldsymbol{e}_v$ is a one-hot vector corresponding to node $v$. During sampling, the probability of selecting a node $u$ at the $k$-hop neighborhood is proportional to its relative importance score:

$$p_k(u) = \frac{\pi_{vu}}{\sum_{w \in \mathcal{N}_k(v)} \pi_{vw}}, \tag{2}$$

where $\mathcal{N}_k(v)$ denotes the set of $k$-hop neighbors of node $v$. Once the sampling is complete, we construct the subgraph by extracting all edges among the selected nodes. It ensures a higher probability of including structurally important nodes while preserving the local neighborhood structure.

**Edge-Level Tasks** For a target edge $(u, v)$, we first apply the node-level subgraph sampling strategy independently to each endpoint, generating two subgraphs $\mathcal{G}_u$ and $\mathcal{G}_v$. The final subgraph representation for the edge is obtained by taking the union of the two:

$$\mathcal{G}_{(u,v)} = \mathcal{G}_u \cup \mathcal{G}_v. \tag{3}$$

This approach effectively captures both the local context around each endpoint and the structural characteristics of the edge itself, making it well-suited for link prediction and other edge-level tasks.

**Graph-Level Tasks**  For graph-level prediction tasks such as molecular property prediction, each graph instance is treated as a complete data sample without additional subgraph sampling. This is because the graph itself already represents a self-contained unit of information.

## 4.2   Knowledge Extraction from LLM

Most existing self-supervised learning methods for graphs adopt GNNs as their backbone architecture and rely on pre-processed node feature vectors as input [22, 28, 29]. However, these approaches often fall short of capturing the rich semantic information embedded within graphs, particularly when dealing with nodes that carry complex textual attributes. As previously discussed, Large Language Models excel at understanding and processing textual information, having been trained on diverse and extensive corpora. This enables them to acquire broad and transferable knowledge for interpreting natural language attributes in graph data. To fully exploit the complementary strengths of structural and textual information, we propose an end-to-end self-supervised learning framework for TAGs. Our method integrates a pre-trained Language Model and a GNN in a cascaded architecture that serves as a teacher model, enabling joint modeling of semantic and structural features. Specifically, we employ Sentence Transformers (ST) [30] as the language model and GCN [15] as the graph encoder. These two components collaboratively capture both the semantic content and topological structure of TAGs.

Inspired by the recent success of masked modeling techniques in natural language processing [31, 32], we design a text-based masked autoencoder framework to enable large-scale self-supervised pretraining on TAGs. By randomly masking portions of node textual attributes and requiring the model to recover the missing content based on contextual and neighborhood information, our approach effectively guides the model to learn latent semantic correlations and structural patterns. This pretraining strategy significantly enhances the model's generalization ability and expressiveness for a variety of downstream tasks.

**Masking Strategy**  During training, each batch processes a (sub)graph $\mathcal{G} = (\mathcal{V}, \mathcal{E}, \mathcal{T}_\mathcal{V})$, where $\mathcal{V}$ denotes the set of nodes, $\mathcal{E}$ the set of edges, and $\mathcal{T}_\mathcal{V}$ the textual features associated with each node. To prepare the textual input, the text of each node is augmented with special tokens: a [CLS] token is added at the beginning to serve as the aggregate representation of the sentence (and thus the node), and a [SEP] token is appended at the end to indicate the end of the sequence.

Let $t_v$ denote the raw textual feature of node $v \in \mathcal{V}$. After tokenization and augmentation, the tokenized input sequence becomes: $t_v = [[\text{CLS}], T_1, T_2, \ldots, T_{n_v}, [\text{SEP}]]$, where $T_i$ are the tokens of the textual input and $n_v$ is the number of tokens for node $v$. To enable self-supervised learning, we apply a token-level masking strategy inspired by masked language modeling. A subset of the tokens in each $t_v$ is randomly selected and replaced with a special [MASK] token. This process is governed by a stochastic masking function $\mathcal{M}(\cdot)$, which determines which positions to mask in each token sequence. Formally, for each token sequence $t_v$, we generate a masked version $\tilde{t}_v$ such that:

$$\tilde{t}_v = \mathcal{M}(t_v) = [[\text{CLS}], \tilde{T}_1, \tilde{T}_2, \ldots, \tilde{T}_{n_v}, [\text{SEP}]], \tag{4}$$

where some $\tilde{T}_i$ are replaced with [MASK] tokens while others remain unchanged. The model is then trained to reconstruct the original tokens at the masked positions based on the surrounding textual context and the structural neighborhood encoded by the GNN. This encourages the model to learn deep semantic representations that are sensitive to both local graph topology and node-specific language attributes.

**Encoder**  The teacher model comprises a language model (LM) $f_{\text{LM}}$ and a graph neural network (GNN) $f_{\text{GNN}}$. For each node $v \in \mathcal{V}$, the masked textual feature sequence $t_v$ is encoded by the LM to obtain hidden representations:

$$\boldsymbol{E}_v = f_{\text{LM}}(\tilde{t}_v), \tag{5}$$

where $\boldsymbol{E}_v \in \mathbb{R}^{(n_v+2)\times d}$ is to the output embeddings of $n_v$ subword tokens along with special tokens such as [CLS] and [SEP]. The embedding of the [CLS] token, denoted by $\boldsymbol{E}^{\text{cls}} \in \mathbb{R}^{|\mathcal{V}|\times d}$, is extracted as the initial representation of nodes. To incorporate structural information, $\boldsymbol{E}^{\text{cls}}$ is propagated through the GNN $f_{\text{GNN}}$ over the adjacency matrix $\boldsymbol{A}$, yielding the fused representation $\boldsymbol{H}^{\text{cls}}$:

$$\boldsymbol{H}^{\text{cls}} = f_{\text{GNN}}(\boldsymbol{A}, \boldsymbol{E}^{\text{cls}}). \tag{6}$$

For each node $v$, we concatenate the textual embedding $\boldsymbol{E}_v$, obtained from the masked forward pass, with the GNN-enhanced [CLS] token representation $\boldsymbol{H}_v^{\text{cls}}$, followed by a linear transformation:

$$\boldsymbol{H}_v = \text{Linear}\left(\boldsymbol{E}_v \oplus \left(\boldsymbol{H}_v^{\text{cls}} \otimes \mathbf{1}_{n_v+2}^\top\right)\right), \tag{7}$$

where $\mathbf{1}_{n_v+2} \in \mathbb{R}^{n_v+2}$ is a column vector of ones. The outer product $\boldsymbol{H}_v^{\mathrm{cls}} \otimes \mathbf{1}_{n_v+2}^{\top}$ replicates the graph-aware node representation across all token positions, which matches the dimensionality of $\boldsymbol{E}_v \in \mathbb{R}^{(n_v+2) \times d}$. The symbol $\oplus$ denotes horizontal concatenation, resulting in a fused representation of shape $(n_v + 2) \times 2d$. The linear layer projects this fused matrix back to the original embedding space: $\mathrm{Linear}(\cdot) : \mathbb{R}^{2d} \to \mathbb{R}^d$. Finally, a language modeling head (MLMHead), implemented as a multi-layer perceptron (MLP), maps the transformed embeddings into the vocabulary space to produce token-level prediction probabilities: $\boldsymbol{P}_v = \mathrm{MLMHead}(\boldsymbol{H}_v)$.

## 4.3 Knowledge Distillation

To enable efficient and scalable deployment, we design a lightweight student model that approximates the teacher's representations while preserving both semantic and structural information. Unlike the teacher model, which relies on explicit message passing, the student model incorporates graph structure implicitly through feature augmentation, thus significantly reducing computational overhead.

The student model adopts a structure-aware multilayer perceptron (MLP) to approximate the teacher's representations. For masked textual feature sequence of each node $\tilde{t}_v$, the input to the student model is constructed by augmenting the [CLS] embedding $\boldsymbol{E}_v^{\mathrm{cls}} \in \mathbb{R}^{1 \times d}$ with its corresponding Personalized PageRank (PPR) scores $p_v \in \mathbb{R}^{1 \times d_p}$ relative to its subgraph neighbors. Specifically, the PPR scores encode the relative importance of neighboring nodes and thereby inject structural information into the input features. The node representation is obtained by applying $f_{\mathrm{MLP}}$ over the concatenated features:

$$\tilde{\boldsymbol{H}}_v = f_{\mathrm{MLP}}\left(\left[\boldsymbol{E}_v^{\mathrm{cls}} \,\|\, p_v\right]\right), \tag{8}$$

where $\|$ represents vector concatenation. By leveraging PPR-based structural priors, the student model can efficiently capture graph topology without relying on explicit message passing, enabling lightweight yet structure-aware representation learning.

**Memory Bank**  To extract representative and diverse features, we introduce a memory bank that stores a set of prototypical representations throughout training. The memory bank comprises $L$ fixed-size memory anchors $\{\boldsymbol{a}_j \in \mathbb{R}^d\}_{j=1}^L$, where each anchor serves as a prototype capturing typical embedding patterns of (sub)graphs.

Given a node $v$, we compute an activation score $s_{vj}$ for each memory anchor $\boldsymbol{a}_j$, which quantifies the similarity between the input embedding and the stored prototypes. Specifically, the memory anchors $\boldsymbol{a}_j$ are initialized from a uniform distribution, following standard embedding initialization practices to ensure stable variance and prevent early model collapse.

During training, the memory anchors are progressively refined through attention-based interactions with incoming graph representations. The activation score $s_{vj}$ is computed as:

$$s_{vj} = \mathcal{S}(\tilde{\boldsymbol{H}}_v, \boldsymbol{a}_j), \tag{9}$$

where $\mathcal{S}(\cdot, \cdot)$ denotes a distance or similarity metric. We then apply a softmax function over the $L$ activation scores to obtain normalized scores $s'_{vj}$:

$$s'_{vj} = \frac{e^{s_{vj}}}{\sum_{k=1}^L e^{s_{vk}}}, \quad \hat{\boldsymbol{H}}_v = \sum_{j=1}^L s'_{vj} \boldsymbol{a}_j, \tag{10}$$

where $\hat{\boldsymbol{H}}_v \in \mathbb{R}^{1 \times d}$ denotes the reconstructed node embedding. The memory bank preserves invariant and semantically meaningful knowledge across training instances. By aligning graph embeddings with prototypical memory anchors, the model is encouraged to focus on stable and consistent features, mitigating overfitting and enhancing generalization to unseen graphs. This mechanism strengthens the model's robustness and predictive capability, particularly in diverse or noisy graph scenarios.

## 4.4 Optimization Objectives

**Mask Loss**  We adopt the Masked Language Modeling (MLM) objective for training. The underlying intuition behind this design is that the model can learn to reconstruct masked tokens of each node's text by leveraging the textual information from its neighboring nodes. This encourages the model to simultaneously understand local semantic content and exploit the structural dependencies

within the graph. The training loss is computed using the cross-entropy loss function for each node $v \in \mathcal{V}$, targeting the prediction of the original tokens at the masked positions. The loss is defined as:

$$\mathcal{L}_{\text{mask}} = -\frac{1}{|\mathcal{V}|} \sum_{v \in \mathcal{V}} \sum_{i=1}^{n_v} \boldsymbol{I}(v, i) \cdot \log \boldsymbol{P}_v[i, T_i], \tag{11}$$

where $|\mathcal{V}|$ is the number of nodes, and $\boldsymbol{I}(v, i)$ is an indicator function, which equals 1 if the $i$-th token in the tokenized text of node $v$ is a [MASK] token and 0 otherwise. $\boldsymbol{P}_v[i, T_i]$ denotes the predicted probability of assigning the ground-truth token $T_i$ to the $i$-th position in the sequence of node $v$, as output by the model. By minimizing this loss, the model is trained to accurately recover masked tokens using the textual context and the structural information encoded in the graph, thereby fostering more informative and robust node representations.

**Consistency Loss** In addition to the masked language modeling (MLM) loss, we further introduce a consistency loss to impose regularization constraints on the latent space, thereby enhancing the stability and alignment of learned representations. The consistency loss consists of two components: one enforces alignment between the student model and the teacher model, while the other maintains consistency with memory-based anchors. For student-teacher consistency, we adopt cosine similarity to encourage the student model (typically a lightweight MLP) to produce embeddings close to those generated by the teacher model (the cascaded LM-GNN architecture). Specifically, given the student representation $\tilde{\boldsymbol{H}}_v$ and the teacher representation $\boldsymbol{H}_v$ for node $v$, the loss is formulated as:

$$\mathcal{L}_{\text{ST}} = 1 - \frac{1}{|\mathcal{V}|} \sum_{v \in \mathcal{V}} \left( 1 - \frac{\boldsymbol{H}_v^T \tilde{\boldsymbol{H}}_v}{\|\boldsymbol{H}_v\| \cdot \|\tilde{\boldsymbol{H}}_v\|} \right). \tag{12}$$

The memory consistency loss enables the model to update the corresponding memory anchors, thereby capturing invariant and prototypical knowledge about generalized graph representations. Specifically, the memory consistency loss is defined as:

$$\mathcal{L}_{\text{ME}} = \frac{1}{|\mathcal{V}|} \sum_{v \in \mathcal{V}} \left\| \hat{\mathbf{H}}_v - \tilde{\mathbf{H}}_v \right\|^2, \tag{13}$$

where $\hat{\mathbf{H}}_v$ denotes the original structure-aware graph embedding for node $v$, and $\tilde{\mathbf{H}}_v$ represents the aligned embedding. By minimizing this loss, the model is encouraged to preserve structural information within the learned representations and refine the memory anchors. This facilitates a more accurate encoding of the essential and invariant characteristics of the graph, thereby enhancing the model's generalization ability across downstream tasks.

The overall loss is then a sum of these three components:

$$\mathcal{L} = \mathcal{L}_{\text{mask}} + \mathcal{L}_{\text{ST}} + \mathcal{L}_{\text{ME}}. \tag{14}$$

By jointly optimizing the MLM loss and the consistency loss, the model is encouraged to capture both semantic and structural information in a stable and generalizable way, leading to more robust node representations for downstream graph tasks.

**Remark 4.1.** *During inference, only the student model is employed to generate node embeddings. Given an unseen (sub)graph and a set of anchor nodes for which we aim to obtain representations, we first use the language model $f_{LM}$ to encode the raw textual attributes of all nodes in the graph. Subsequently, the [CLS] tokens from each node are passed into the MLP module $f_{MLP}$ to produce propagated representations, which serve as the final embeddings for each node. Finally, we extract the embeddings corresponding to the specified anchor nodes for downstream use.*

## 5 Experiments

### 5.1 Experimental Setting

We adopt the widely used linear probing protocol to evaluate the representation learning capability of the self-supervised pretraining models on unseen datasets. Specifically, we train a linear classifier on top of the frozen embeddings produced by the pre-trained models. Both our model and all baseline

Table 1: Experimental results for self-supervised representation learning. We report the accuracy ( % ) for the node classification task and the ROC - AUC score ( % ) for the link prediction task. The proposed method and other self-supervised benchmarks are pretrained on ogbn - Paper100M and then evaluated on individual target datasets. The best results are **bold**, and the second best are underlined.

| | Node Classification (Accuracy, %) | | | | | Link Prediction (ROC-AUC, %) | | |
| --- | --- | --- | --- | --- | --- | --- | --- | --- |
| | Cora | Pubmed | ogbn-Arxiv | WikiCS | Products | FB15K237 | WN18RR | ML1M |
| GCN | 57.62 ± 0.21 | 55.18 ± 0.37 | 60.85 ± 0.13 | 53.24 ± 0.23 | 61.95 ± 0.32 | 72.52 ± 0.29 | 72.05 ± 0.31 | 66.64 ± 0.52 |
| GIN | 57.97 ± 0.45 | 48.98 ± 0.21 | 61.27 ± 0.25 | 52.32 ± 0.36 | 63.83 ± 0.15 | 73.60 ± 0.33 | 73.98 ± 0.39 | 65.71 ± 0.37 |
| GAT | 66.29 ± 0.24 | 57.30 ± 0.35 | 63.34 ± 0.49 | 50.91 ± 0.34 | 64.94 ± 0.28 | 72.14 ± 0.43 | 72.57 ± 0.65 | 66.89 ± 0.34 |
| GraphCL | 72.56 ± 0.52 | 67.27 ± 1.21 | 62.15 ± 0.21 | 55.96 ± 1.02 | 72.18 ± 0.42 | 65.34 ± 0.87 | 68.52 ± 0.55 | 67.02 ± 0.49 |
| BGRL | 74.42 ± 0.81 | 68.17 ± 0.22 | 69.04 ± 0.14 | 59.93 ± 0.35 | 73.08 ± 0.28 | 64.92 ± 0.36 | 66.47 ± 0.43 | 68.10 ± 0.22 |
| GraphMAE | 73.54 ± 0.38 | 68.38 ± 1.18 | 68.54 ± 0.20 | 54.68 ± 0.55 | 72.65 ± 0.62 | 62.87 ± 0.84 | 70.51 ± 0.32 | 68.57 ± 0.34 |
| GraphMAE2 | 73.92 ± 0.64 | 68.76 ± 0.55 | 69.07 ± 0.27 | 58.04 ± 0.47 | 74.05 ± 0.33 | 60.54 ± 0.39 | 71.43 ± 0.11 | 69.13 ± 1.01 |
| Graph-LLM | 73.88 ± 0.35 | 68.62 ± 0.32 | 70.11 ± 0.52 | 62.16 ± 0.48 | 74.02 ± 0.34 | 82.47 ± 0.56 | 73.46 ± 0.61 | 70.21 ± 0.51 |
| UniGraph | 74.65 ± 0.56 | 70.84 ± 0.51 | 70.89 ± 0.44 | 65.47 ± 0.51 | 76.58 ± 0.44 | 85.01 ± 0.63 | 80.55 ± 0.27 | 70.02 ± 0.28 |
| **SSTAG (Ours)** | **75.09 ± 1.02** | **72.65 ± 0.35** | **72.85 ± 0.43** | **68.76 ± 0.62** | **78.27 ± 0.48** | **88.64 ± 0.49** | **82.42 ± 0.66** | **71.24 ± 0.42** |

Table 2: Experimental results for self-supervised representation learning. We report the ROC - AUC ( % ) for the graph classification task and RMSE ( $\Downarrow$ ) for the graph regression task. " $\Downarrow$ " indicates that lower RMSE values correspond to better model performance. SSTAG and other self-supervised benchmarks are pretrained on ogbn-Paper100M and then evaluated on individual target datasets.

| | Graph Classification (ROC-AUC, %) | | | | Graph Regression (RMSE, $\Downarrow$) | | |
| --- | --- | --- | --- | --- | --- | --- | --- |
| | HIV | BBBP | BACE | MUV | esol | LIPO | CEP |
| GCN | 74.15 ± 0.26 | 65.43 ± 0.33 | 69.02 ± 0.38 | 71.82 ± 0.26 | 1.379 ± 0.034 | 0.824 ± 0.034 | 1.342 ± 0.036 |
| GIN | 74.38 ± 0.24 | 66.07 ± 0.52 | 69.85 ± 0.32 | 72.35 ± 0.14 | 1.295 ± 0.021 | 0.819 ± 0.021 | 1.296 ± 0.015 |
| GAT | 73.82 ± 0.43 | 66.82 ± 0.15 | 68.51 ± 0.20 | 72.06 ± 0.42 | 1.324 ± 0.027 | 0.821± 0.027 | 1.305 ± 0.008 |
| GraphCL | 75.55 ± 0.29 | 68.74 ± 0.38 | 73.64 ± 0.56 | 74.27 ± 0.37 | 1.304 ± 0.024 | 0.763 ± 0.024 | 1.326 ± 0.016 |
| BGRL | 75.32 ± 0.44 | 67.35 ± 0.42 | 75.14 ± 0.21 | 75.13 ± 0.35 | 1.162 ± 0.018 | 0.784 ± 0.018 | 1.293 ± 0.021 |
| GraphMAE | 76.13 ± 0.12 | 69.51 ± 0.14 | 76.28 ± 0.43 | 75.88 ± 0.26 | 1.116 ± 0.015 | 0.754 ± 0.015 | 1.288 ± 0.008 |
| GraphMAE2 | 77.84 ± 0.35 | 71.62 ± 0.25 | 77.41 ± 0.18 | 77.69 ± 0.42 | 1.069 ± 0.006 | 0.728 ± 0.006 | 1.262 ± 0.011 |
| Graph-LLM | 76.43 ± 0.20 | 72.54 ± 0.37 | 80.65 ± 0.33 | 76.13 ± 0.31 | 1.114 ± 0.024 | 0.719 ± 0.024 | 1.232 ± 0.009 |
| UniGraph | 77.27 ± 0.31 | 73.28 ± 0.30 | 79.23 ± 0.26 | 76.88 ± 0.52 | 1.090 ± 0.032 | 0.710 ± 0.032 | 1.195 ± 0.012 |
| **SSTAG (Ours)** | **79.52 ± 0.26** | **74.38 ± 0.35** | **82.06 ± 0.31** | **79.86 ± 0.40** | **1.043 ± 0.020** | **0.698 ± 0.003** | **1.186 ± 0.006** |

self-supervised methods are first pre-trained on the large-scale citation network ogbn-Paper100M. Subsequently, we evaluate the learned representations on twelve graph datasets spanning five distinct domains. For baselines, we compare our method with the state-of-the-art generative self-supervised methods for graphs: GraphMAE [22] and GraphMAE2 [33], contrastive methods such as GraphCL [20] and BGRL [34], and methods specifically tailored for TAGs, including UniGraph [35] and Graph-LLM [9]. Since most of these baselines are not originally designed for cross-domain evaluation, we use the ST language model to unify the input node features across different graphs. To ensure a fair comparison, all baselines employ GCN as the backbone GNN, consistent with our method. Detailed descriptions of the datasets, baselines and hyperparameter settings can be found in Appendix A.

## 5.2 Self-Supervised Representation Learning

We comprehensively evaluate the proposed SSTAG framework on four graph representative tasks: node classification, link prediction, graph classification, and graph regression. The overall experimental results are summarized in Tables 1 and 2, from which three key observations can be drawn. (1) Across all tasks and datasets, SSTAG consistently outperforms a wide range of existing graph self-supervised learning methods. This highlights its strong generalization capability in cross-domain graph learning scenarios, where data distributions and structural patterns vary significantly. By capturing richer structural and semantic dependencies, SSTAG generates more discriminative and transferable graph embeddings, enabling it to adapt effectively to unseen graphs. (2) As a standalone pretraining model, SSTAG exhibits remarkable performance when transferred to downstream tasks. In most cases, it achieves results comparable to, or even better than, fully supervised baselines, particularly in low-label regimes where annotated data is scarce. For instance, on the BACE graph classification dataset, SSTAG achieves an accuracy of 82.06% after fine-tuning, which surpasses the supervised baseline by 12.21%. This clearly demonstrates the label efficiency of our method, underscoring its potential in real-world scenarios where labeled graph data is often limited or costly to obtain. (3) Another important finding is that SSTAG benefits from a unified task template, which allows it to seamlessly adapt to tasks of different granularities, ranging from local node-level tasks to holistic graph-level tasks. This flexible adaptation mechanism contributes to its robust performance

Table 3: Ablation studies of key components.

|  | WikiCS | ogbn-Arxiv | FB15K237 | MUV |
|---|---|---|---|---|
| SSTAG | 68.76 | 72.85 | 88.64 | 79.86 |
| W/o $\mathcal{L}_{mask}$ | 67.02 | 70.51 | 85.84 | 76.22 |
| W/o $\mathcal{L}_{ST}$ | 67.75 | 71.86 | 87.12 | 78.65 |
| W/o $\mathcal{L}_{ME}$ | 66.53 | 71.14 | 85.96 | 76.43 |
| W/o GNN | 64.34 | 69.53 | 84.32 | 70.57 |
| W/o PPR | 68.12 | 72.37 | 88.4 | 79.21 |

Table 4: Analysis of distillation and inference efficiency on ogbn-Arxiv.

| Model | Inference Time (min) | Accuracy (%) | Parameters |
|---|---|---|---|
| GNN | 13.4 | 73.91 | 7.1B |
| Distilled MLP | 8.7 | 72.85 | 22M |
| $\Delta$ Change | $\downarrow 35.1\%$ | $\downarrow 1.06\%$ | $\downarrow 99.7\%$ |

Table 5: Analysis of LMs and GNNs choices.

|  | #parameters | ogbn-Arxiv | FB15K237 | MUV |
|---|---|---|---|---|
| Sentence Transformer [30] | $\sim 66M$ | 72.85 | 88.64 | 79.86 |
| DeBERTa-v3-base [36] | $\sim 184M$ | 72.53 | 88.83 | 79.54 |
| E5-large-v2 [37] | $\sim 335M$ | 73.21 | 89.02 | 80.01 |
| LLaMA-2-7B-hf [38] | $\sim 7B$ | 73.68 | 89.67 | 80.39 |
| GCN | − | 72.85 | 88.64 | 79.86 |
| GIN | − | 72.43 | 89.13 | 80.04 |
| GAT | − | 73.02 | 88.92 | 80.33 |

in complex multi-task settings, where heterogeneous objectives must be addressed simultaneously. The ability to transfer knowledge across tasks and granularities further underscores the scalability and versatility of the proposed framework.

## 5.3 Ablation Studies

**Ablation on Key Components** We perform an ablation study to evaluate the contribution of each component in SSTAG, with results summarized in Table 3. The variant "W/o $\mathcal{L}_{mask}$" removes the masked modeling objective and trains the model only with the consistency loss, and the performance drop highlights the role of masked modeling in capturing fine-grained structural and semantic dependencies. The variant "W/o $\mathcal{L}_{ST}$" discards the student–teacher consistency while retaining masked modeling and memory-based consistency, and the degradation indicates that the student–teacher design is crucial for stabilizing training and improving cross-view alignment. In contrast, "W/o $\mathcal{L}_{ME}$" eliminates memory-based consistency while keeping masked modeling and student–teacher consistency, and the observed decline suggests that memory-based consistency is effective in preserving long-range dependencies and mitigating representation drift. The setting "W/o GNN" replaces the graph neural network encoder with a standard MLM objective for fine-tuning the language model, followed by distillation into an MLP, and the substantial performance loss demonstrates the necessity of graph-structured message passing for relational reasoning. Finally, "W/o PPR Sampling" substitutes personalized PageRank-based sampling with simple neighborhood sampling, and the reduction confirms that PPR-based sampling provides more informative subgraph contexts and alleviates bias from naive neighborhood expansion. Overall, the consistent degradation across all variants demonstrates that each component is indispensable, and their joint design is critical to the effectiveness and robustness of SSTAG.

**Impact of Model Distillation** We further evaluate the effectiveness and efficiency of the distilled MLP model in comparison to the original GNN teacher. Specifically, we measure inference accuracy, computational efficiency, and model size. The distilled MLP achieves a 35.1% improvement in inference efficiency and a 99.7% reduction in the number of parameters, while incurring only a minor accuracy drop of 1.06% relative to the GNN. These results demonstrate that our distillation approach preserves the representational power of the teacher model while substantially reducing computational and memory overhead. As shown in Table 4, the distilled MLP consistently maintains high accuracy across multiple downstream tasks, confirming that the co-distillation strategy effectively transfers knowledge from the GNN to the lightweight student model without sacrificing task performance.

**Analysis of LMs and GNNs Choices** Table 5 presents a comparative analysis of different language models (LMs) and GNNs as backbone architectures. To assess the impact of LMs, we evaluate several widely used pre-trained models. Compared with SentenceTransformers (ST, 110M parameters), larger models such as E5-large-v2 (335M) [37] and LLaMA-2-7B-hf (7B) [38] yield consistent improvements. For instance, replacing ST with LLaMA-2-7B-hf on the `ogbn-Arxiv` node classifica-

Table 6: Comparison of computational cost and performance of different methods.

| Dataset | Method | Pre-training | Downstream Training | Downstream Inference | Accuracy(%) |
|---------|--------|-------------|---------------------|---------------------|-------------|
| ogbn-Arxiv | GAT | – | 24.6mins | 5.8min | 63.34 |
| | GraphCL | – | 32.6mins | 4.9min | 62.15 |
| | GraphMAE2 | – | 5.2h | 5.1min | 68.76 |
| | Graph-LLM | 24.2h | – | 12.6min | 72.85 |
| | **SSTAG (Ours)** | 22.6h | – | 8.7min | 72.85 |

tion task improves accuracy by 0.82%, highlighting the benefit of higher-capacity LMs. However, these gains incur significantly higher computational and memory costs, revealing an inherent trade-off between accuracy and efficiency. Hence, LM selection should balance task requirements and available resources: smaller models suit low-latency or resource-limited settings, while larger ones favor accuracy-oriented applications. For GNNs, models with stronger aggregation capability, such as GraphSAGE and GAT, generally outperform simpler architectures like GCN, indicating that expressive structural encoders complement high-capacity LMs in enhancing downstream performance. Further LM configuration details are provided in Appendix A.

## 5.4 Efficiency Analysis

The overall time complexity of the proposed method is primarily dominated by the language model (LM), owing to its long-sequence processing. During pretraining, the computational cost is approximately $\mathcal{O}(N \cdot (L^2 d + L_t d^2))$, where $N$ denotes the number of nodes, $L_t$ the input sequence length, and $d$ the embedding dimension. The neighborhood aggregation in the graph neural network (GNN) introduces an additional overhead of $\mathcal{O}(Nd^2 + Ed)$, where $E$ is the number of edges; in dense graphs ($E \propto N^2$), this can grow to $\mathcal{O}(N^2 d)$. In the student model, explicit message passing is replaced by structure-aware MLPs with PPR-based feature injection, thereby reducing the complexity to $\mathcal{O}(Nd)$. Memory retrieval incurs an additional cost of $\mathcal{O}(NLd)$, where $L$ is the number of memory anchors. Other components, such as masked prediction in multimodal interaction ($\mathcal{O}(n_{\text{masked}} \cdot n_v)$, where $n_{\text{masked}}$ is the number of masked tokens) and consistency loss ($\mathcal{O}(Nd)$), introduce relatively minor computational overhead. Overall, the dominant factor remains the LM complexity, while the student model and auxiliary modules are designed to maintain scalability in large-scale graph settings.

As shown in Table 6, the training time and memory overhead of SSTAG are comparable to those of training a language model (LM) using only the masked language modeling (MLM) objective. This suggests that the overall computational cost of our framework is primarily dominated by the LM. Consequently, when using similar LMs, the runtime of SSTAG is on par with other LM-based approaches. SSTAG is designed as a pretraining-centric model, where most of the computational cost is incurred during the pretraining phase. However, it offers a key advantage at inference time by allowing the use of a distilled student model (structure-aware MLP) resulting in significantly lower inference overhead. We further compare the training and inference costs of our model with GNN-based methods. We conduct experiments on two datasets of different scales: ogbn-arXiv and WikiCS. Although SSTAG incurs longer pretraining time, its inference time on downstream datasets is comparable to or even shorter than the combined training and inference time of GNN-based methods. This advantage becomes more pronounced as the size and number of downstream datasets increase. While LMs generally have larger parameter counts, our framework mitigates this drawback by requiring only forward passes during downstream inference, thereby avoiding the additional memory overhead of backpropagation during training.

## 6 Conclusion

In this work, we propose **SSTAG**, a structure-aware self-supervised framework tailored for text-attributed graphs, aiming to bridge the gap between the structural reasoning strengths of GNNs and the semantic understanding capabilities of LLMs. By leveraging text as a unified medium, SSTAG tackles the challenge of knowledge transfer across heterogeneous graph domains. Our approach introduces a generic prediction template for node-, edge-, and graph-level tasks, along with a novel co-distillation objective that fuses multimodal knowledge into a lightweight, structure-aware MLP. Extensive experiments demonstrate that SSTAG not only achieves superior performance across cross-domain and large-scale settings but also substantially reduces inference costs, making it a promising direction for practical and scalable graph representation learning.

## Acknowledgment

This work is supported in part by the National Natural Science Foundation of China (No.62106259, No.62076234), Beijing Outstanding Young Scientist Program (NO.BJJWZYJH012019100020098), and Beijing Natural Science Foundation (No. 4222029).

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

# A   Details of Experiments

The supplementary material provides additional details on the experiments section that could not be included in the main manuscript due to page limitations.All experiments were conducted on a Linux server equipped with 945GB of RAM and eight NVIDIA A100 GPUs, each with 40GB of memory. The implementation of our method is available at `https://github.com/Liury925/SSTAG`.

## A.1   Datasets

In this section, we describe the datasets used in this work. The overall statistics for each dataset are given in Table 7.

**Cora**   The Cora [9] dataset represents a co-citation graph of academic papers in the field of computer science. In *Graph-LLM*, the authors reconstruct this dataset because the commonly used Cora version in the GNN community relies on bag-of-words features, making it difficult to retrieve the original text. The newly collected Cora dataset contains 2,708 nodes and 10,556 edges, maintaining the same graph structure as the original version.

**PubMed**   The PubMed [26] dataset is a co-citation graph of biomedical research papers focused on diabetes mellitus. The data source and processing procedure follow the same approach as the Cora dataset. After preprocessing, the dataset contains 19,717 nodes and 88,648 edges. For the node classification task, nodes are categorized into three classes: *Diabetes mellitus, experimental*, *Diabetes mellitus, type 1*, and *Diabetes mellitus, type 2*. The standard train/validation/test split consists of 60 training nodes, 500 validation nodes, and 19,157 test nodes.

**ogbn-Arxiv**   The Arxiv [39] dataset is a large-scale citation graph constructed from academic papers published on the arXiv platform. The graph comprises 169,343 nodes and 1,166,243 edges. It is primarily used for the node classification task, where each node corresponds to a paper, and edges represent citation relationships. The dataset includes a total of 40 distinct classes. The standard data split contains 90,941 training, 29,799 validation, and 48,603 test nodes.

**ogbn-Papers100M**   The ogbn-Papers100M [39] dataset is part of the Open Graph Benchmark (OGB) and contains over 111 million nodes and 1.6 billion edges. Each node represents a paper from the Microsoft Academic Graph, and edges denote citation relationships. The task is node classification, where the goal is to predict the field of study for each paper. Due to its massive scale, the dataset is designed to evaluate the scalability and efficiency of graph learning algorithms.

**WikiCS**   The WikiCS [26] is a graph dataset constructed from the English Wikipedia, where nodes correspond to articles and edges represent hyperlink connections. Each article is associated with textual features and is labeled by one of several pre-defined classes. The task is semi-supervised node classification, and it includes 10 different training/validation/test splits, allowing for robust evaluation under few-shot settings.

**Products**   The Products [39] dataset is part of the Amazon co-purchase graph, where nodes are products and edges connect products frequently bought together. It is included in the OGB benchmark as ogbn-products. Each node is associated with a multi-hot encoded feature vector and a category label. The dataset is used for node classification, with over 2 million nodes and 60+ classes.

**FB15K237**   FB15k237 [26] is a commonly used benchmark in knowledge graph completion tasks. It is a refined version of the original FB15k dataset, which was curated from Freebase. The refinement removes inverse relations to avoid test leakage. The dataset includes entities as nodes and relations as labeled edges, and the primary task is link prediction or knowledge graph completion.

**WN18RR**   WN18RR [26] is a benchmark knowledge graph dataset derived from WordNet. It is a variant of WN18 with inverse relations removed to prevent test leakage. The graph consists of entities and labeled edges representing lexical relationships such as hypernymy and synonymy. It is widely used for evaluating link prediction models in knowledge graphs.

Table 7: Statistics of text-attributed graph datasets.

| Dataset | Avg. #N | Avg. #E | #G | Task level | Task(class) | Domain | Split (train/val/test) |
|---|---|---|---|---|---|---|---|
| Cora | 2,708 | 10,556 | 1 | Node | classification(7) | Citation | 140/500/2,068 |
| Pubmed | 19,717 | 88,648 | 1 | Node | classification(3) | Citation | 60/500/19,157 |
| ogbn-Arxiv | 169,343 | 1,166,243 | 1 | Node | classification(40) | Citation | 90,941/29,799/48,603 |
| ogbn-Papers100M | 111,059,956 | 1,615,685,872 | 1 | Node | classification(172) | Citation | 1,196,087/125,265/214,326 |
| WikiCS | 11,701 | 216,123 | 1 | Node | classification(10) | Web link | 580/1,769/5,847 |
| Products | 54,025 | 144,638 | 1 | Node | classification(47) | Co-purchase | 14,695/1,567/36,982 |
| fb15k237 | 14,541 | 310,116 | 1 | Link | classification(237) | Knowledge | 272,115/17,535/20,466 |
| WN18RR | 40,943 | 93,003 | 1 | Link | classification(11) | Knowledge | 86,835/3,034/3,134 |
| ML1M | 9,923 | 2,000,418 | 1 | Link | classification(5) | Movie rating | 850,177/50,011/100,021 |
| HIV | 25.51 | 54.94 | 41,127 | Graph | classification(2) | molecular | 32,901/4,113/4,113 |
| BBBP | 24.06 | 51.91 | 2,039 | Graph | classification(2) | molecular | 1,631/204/204 |
| BACE | 34.09 | 73.72 | 1,513 | Graph | classification(2) | molecular | 1,210/151/152 |
| MUV | 24.23 | 52.56 | 93,087 | Graph | classification(17) | molecular | 74,469/9,309/9,309 |
| ESOL | 13.29 | 27.35 | 1,128 | Graph | Regression | molecular | 902/113/113 |
| CEP | 38.02 | 41.00 | 29978 | Graph | Regression | molecular | 23,982/2,998/2,998 |
| LIPO | 27.04 | 59.00 | 4,200 | Graph | Regression | molecular | 3,360/420/420 |

**ML1M**   The MovieLens-1M [40] dataset is a widely used benchmark for recommender systems. It can be represented as a bipartite user-item interaction graph. Node features include user and item attributes such as age, gender, occupation, and genres. The typical task is rating prediction or top-k recommendation.

**HIV**   The HIV [41] dataset is a molecular graph classification dataset from the MoleculeNet benchmark. Each molecule is represented as a graph, where atoms are nodes and bonds are edges. The binary classification task is to predict whether a molecule is active against HIV. The dataset is used to evaluate models in molecular property prediction.

**BBBP**   The BBBP [41] dataset is a binary classification dataset that predicts whether a given compound can penetrate the blood–brain barrier. Each data point is a molecular graph with atom-level features. This dataset is particularly relevant for drug discovery applications and poses a challenge due to its relatively small size and imbalanced labels.

**BACE**   The BACE [41] dataset contains molecular graphs used to predict the binding results of human $\beta$-secretase 1 (BACE-1) inhibitors. It is a binary classification task that plays a role in early-stage drug development, especially for Alzheimer's disease. Graph-based models leverage atom and bond features to make predictions.

**MUV**   The MUV (Maximum Unbiased Validation) [41] dataset is designed to serve as a challenging benchmark for virtual screening. It includes a collection of molecular graphs with multiple binary classification tasks, each corresponding to a biological target. The dataset is highly imbalanced and contains a significant number of decoys, making it suitable for testing model robustness.

**ESOL**   The ESOL [41] dataset is used for regression tasks where the goal is to predict the aqueous solubility of compounds. Molecules are represented as graphs, and the target is a continuous solubility value. This dataset is important for evaluating models in pharmaceutical and materials chemistry.

**CEP**   The CEP (Clean Energy Project) [41] dataset comprises molecular graphs of organic photovoltaic compounds. Each molecule has a computed power conversion efficiency (PCE), making the task a regression problem. It is one of the largest publicly available molecular property datasets and is critical for materials discovery in renewable energy research.

**LIPO**   The LIPO [41] dataset is a molecular property prediction dataset where the target is the logarithm of the partition coefficient between octanol and water (logP), reflecting the molecule's lipophilicity. It is a regression dataset used in computational chemistry and drug design, where accurate logP prediction is essential for pharmacokinetics modeling.

## A.2   Baselines

**GCN**   Graph Convolutional Network (GCN) [15] introduces convolutional operations into graph-structured data. It aggregates features from a node's neighbors and itself, enabling effective semi-

supervised learning on graph data. GCN is widely used in node classification tasks and serves as the backbone for many subsequent GNN models.
**Code:** `https://github.com/tkipf/gcn`

**GIN**    Graph Isomorphism Network (GIN) [42] is designed to have maximum expressive power among GNNs, equivalent to the Weisfeiler-Lehman test for graph isomorphism. By using a summation-based aggregation and MLP update, GIN can effectively distinguish different graph structures.
**Code:** `https://github.com/weihua916/powerful-gnns`

**GAT**    Graph Attention Network (GAT) [43] applies attention mechanisms to assign different importances to different neighbors during message passing. This allows the model to better capture local structural variations and learn more robust node embeddings.
**Code:** `https://github.com/PetarV-/GAT`

**GraphCL**    GraphCL [20] is a contrastive self-supervised learning framework for graphs. It generates multiple augmented graph views via structural and attribute perturbations and maximizes agreement between their representations. GraphCL has shown competitive performance in unsupervised graph classification.
**Code:** `https://github.com/Shen-Lab/GraphCL`

**BGRL**    BGRL [34] is a bootstrap-based self-supervised method that eliminates the need for negative samples. Inspired by BYOL, it uses two networks—an online and a target network—to predict node embeddings across augmented views. This method is memory-efficient and stable on large graphs.
**Code:** `https://github.com/nerdslab/bgrl`

**GraphMAE**    GraphMAE [22] is a masked autoencoder designed for graphs, inspired by BERT-style pretraining. It masks parts of node features and learns to reconstruct them using a GNN backbone, enabling effective pretraining for downstream tasks.
**Code:** `https://github.com/THUDM/GraphMAE`

**GraphMAE2**    GraphMAE2 [33] is an enhanced version of GraphMAE with improved masking strategies and decoder designs. Using GAT as the encoder, it introduces better training stability and performance in graph representation learning.
**Code:** `https://github.com/THUDM/GraphMAE-v2`

**Graph-LLM**    Graph-LLM [9] is a framework designed to bridge graph representation learning and large language models (LLMs). It introduces a graph-to-text conversion pipeline that transforms graph-structured data into natural language sequences, enabling pretrained LLMs to reason over and extract knowledge from graphs. Graph-LLM supports both node-level and graph-level tasks by prompting the LLMs with rich textual contexts that reflect topological and semantic information. This approach bypasses the need for message passing in traditional GNNs, offering a scalable alternative for graph-based learning.
**Code:** `https://github.com/CurryTang/Graph-LLM.`

**UniGraph**    UniGraph [35] proposes a unified pretraining framework for graph-level, node-level, and edge-level tasks. By designing a universal contrastive learning objective and architecture, UniGraph generalizes well across diverse graph tasks.
**Code:** `https://github.com/Graph-COM/UniGraph`

### A.3   Hyperparameter Setting

The hyperparameter tuning process in this work is divided into three categories. First, some hyperparameters (such as the number of epochs, learning rate, optimizer, and batch size) are selected empirically based on standard practice. Second, certain hyperparameters (such as the masking rate and the number of memory anchors) are optimized using grid search on the validation split, with metrics like accuracy or AUC depending on the specific task. Third, for parameters with complex interactions, we select the best values based on cross-validation across multiple settings. The selection

criteria and the actual values used in our experiments are summarized in Table 8. It is worth noting that optimal values may vary slightly across different datasets.

Table 8: Summary of hyperparameters and tuning criteria.

| Hyperparameter | Value | Tuning Criterion (Search Range) |
|---|---|---|
| Mask Rate | 0.5 | Grid Search (0.1, 0.3, 0.5, 0.7) |
| Num GNN Layers | 3 | Empirical |
| Hidden Size | 768 | Empirical |
| PPR Top-k | 128 | Cross-validation (64, 128, 256) |
| Learning Rate | 2e-5 | Empirical |
| PPR $\alpha$ | 0.15 | Cross-validation (0.10, 0.15, 0.20, 0.25) |
| Weight Decay | 0.001 | Empirical |
| Batch Size | 1024 | Empirical |
| Dropout | 0.2 | Grid Search (0.1, 0.2, 0.3, 0.4) |
| Optimizer | AdamW | Empirical |
| Num Epochs | 1 | Empirical |
| Warmup Steps | 10% | Empirical |
| Num MLP Layers | 3 | Empirical |
| Memory Anchors | 256 | Grid Search (64, 128, 256, 512) |

## A.4 Language Models

**Sentence Transformer** Sentence Transformers [30] are a family of models that extend pretrained transformers like BERT to generate semantically meaningful sentence embeddings. By fine-tuning on natural language inference and paraphrase datasets using Siamese or triplet networks, Sentence Transformers enable efficient semantic similarity search, clustering, and information retrieval.
**HuggingFace:** `https://huggingface.co/sentence-transformers`

**DeBERTa-v3-base** DeBERTa [36] improves BERT and RoBERTa by disentangling the representation of content and position, and using an enhanced mask decoder. The v3 version incorporates further improvements such as better initialization and larger-scale training. DeBERTa-v3-base has around 140M parameters and achieves strong performance on various NLU benchmarks.
**HuggingFace:** `https://huggingface.co/microsoft/deberta-v3-base`

**E5-large-v2** E5-large-v2 [37] (Embedding-from-Embedding) is a dual-encoder model developed by Google for high-quality semantic search and retrieval tasks. It is fine-tuned on a mixture of supervised and unsupervised datasets with contrastive loss to produce universal embeddings. The "large-v2" version contains approximately 355M parameters and supports both query and passage encoding.
**HuggingFace:** `https://huggingface.co/intfloat/e5-large-v2`

**LLaMA-2-7B-hf** LLaMA 2 [38] is a series of open foundation language models released by Meta. LLaMA-2-7B-hf is the 7-billion-parameter variant and is suitable for a wide range of NLP tasks, including generation, question answering, and dialogue. The Hugging Face version provides easy integration with the Transformers library.
**HuggingFace:** `https://huggingface.co/meta-llama/Llama-2-7b-hf`

