# OpenReview forum: "SSTAG: Structure-Aware Self-Supervised Learning Method for Text-Attributed Graphs"
_NeurIPS.cc/2025/Conference — NeurIPS 2025 poster_

### Official Review · Reviewer_MtjL · 2025-06-28

**Clarity:** 3
**Significance:** 3
**Originality:** 3
**Rating:** 4
**Confidence:** 3

**Summary:**

The paper introduces a novel self-supervised learning framework called SSTAG for text-attributed graphs that leverages large language models and graph neural networks to create a unified representation. It aims to bridge the gap between semantic reasoning and structural modeling by using text as a common medium. The framework includes a dual knowledge distillation process that transfers knowledge from LLMs and GNNs into a lightweight, structure-aware MLP, enhancing scalability and reducing inference costs. Additionally, it incorporates an in-memory mechanism to store typical graph representations, improving generalization across different domains. Extensive experiments demonstrate its superior performance in cross-domain transfer learning and scalability.

**Questions:**

1. The authors use the PPR algorithm, which involves matrix inversion. The sampled subgraph has quadratic complexity.
2. What kind of interpretable advantages can the graph + LLM framework provide?
3. The authors use a consistency loss instead of a contrastive loss. What is the motivation behind this choice, and are there experiments to verify its advantages?
4. The authors have demonstrated the superiority of their framework on various tasks, but it seems that they haven't tried the most common graph clustering task.
5. While LLM-based ideas are well motivated, the mechanism could be further explained in terms of semantic alignment between textual and graph domains.
6. There are a few minor grammatical errors in the paper. Please check them carefully.

**Ethical Concerns:**

["NO or VERY MINOR ethics concerns only"]

**Final Justification:**

The author's response has essentially resolved my main concerns. I hope the author can incorporate the additional experiments and clearer explanations into the future revised version. Given this, I will maintain my positive score.

**Limitations:**

Yes

**Paper Formatting Concerns:**

There are no major formatting issues.

**Quality:**

3

**Strengths And Weaknesses:**

Strengths
1. The authors propose a novel framework for the task of OOD detection on graph data, which is highly practical and likely to attract broad interest.
2. The paper is well-structured and easy to follow, making it accessible for readers with limited background knowledge.
3. The authors demonstrate superior performance over state-of-the-art methods across diverse datasets.
4. The authors conduct thorough experiments to validate the proposed approach's effectiveness.

Weaknesses
1. The authors use the PPR algorithm, which involves matrix inversion. The sampled subgraph has quadratic complexity.
2. What kind of interpretable advantages can the graph + LLM framework provide?
3. The authors use a consistency loss instead of a contrastive loss. What is the motivation behind this choice, and are there experiments to verify its advantages?
4. The authors have demonstrated the superiority of their framework on various tasks, but it seems that they haven't tried the most common graph clustering task.
5. While LLM-based ideas are well motivated, the mechanism could be further explained in terms of semantic alignment between textual and graph domains.
6. There are a few minor grammatical errors in the paper. Please check them carefully.

---

> ### Author Rebuttal · Authors · 2025-07-31
>
> We sincerely thank the reviewer for the thoughtful comments and constructive suggestions. We provide our feedback as follows.
>
> >**`Q1`: The authors use the PPR algorithm, which involves matrix inversion. The sampled subgraph has quadratic complexity.**
>
> **AQ1**: Thank you for your questions. In practice, we **do not perform matrix inversion on the entire large graph**. Instead, during subgraph sampling, we iteratively compute the local neighborhood, which greatly **reduces the algorithm’s complexity**. The actual computational cost mainly depends on the number of sampled nodes and the number of iterations, rather than the square of the total number of nodes in the graph.
>
> Additionally, we compared the training and inference times of SSTAG with other methods on large-scale datasets. As shown in Table 5, our model uses only a structure-aware MLP for inference, which significantly reduces computational overhead and ensures scalability in real-world applications.
>
> >**`Q2`: What kind of interpretable advantages can the graph + LLM framework provide?**
>
> **AQ2**: The integration of GNNs and LLMs in SSTAG offers several interpretability benefits:
>
> - **Semantic-Structural Fusion**: GNNs encode local graph topology, while LLMs embed high-level semantic features from textual attributes. Our framework enables interpretability both in terms of which neighboring nodes contribute to predictions (via structure), and what semantic features are being captured (via language).
> - **Modality Alignment via Memory Anchors**: The proposed memory bank mechanism aligns subgraph embeddings with prototypical semantic-structural anchors, facilitating a clearer understanding of what type of substructure or semantic pattern the model relies on.
> - **Unified Embedding Space**: Our unified architecture supports representation sharing across node-, edge-, and graph-level tasks, making it easier to track embedding behaviors across different prediction granularities.
>
> >**`Q3`: The authors use a consistency loss instead of a contrastive loss. What is the motivation behind this choice, and are there experiments to verify its advantages?**
>
> **AQ3**: We chose to use **consistency loss** instead of traditional contrastive loss for several reasons:
>
> - **Teacher-Student Distillation:**
>   The training paradigm uses a cascaded LM+GNN teacher and a structure-aware MLP student. In this setting, directly aligning the embeddings between the teacher and the student via consistency loss is **simpler** and **more stable** than generating positive and negative pairs required by contrastive loss. Moreover, most knowledge distillation works also employ consistency-based objectives for student-teacher alignment [1, 2].
>
> - **Memory Consistency:**
>   In addition to aligning the student with the teacher, we further encourage consistency between the student and the prototype memory anchors. This extends beyond typical contrastive learning frameworks by leveraging memory-based regularization.
>
> - **Empirical Evidence:**
> As shown in **Table 3**, removing either the student-teacher or memory-based consistency loss leads to a significant performance drop, highlighting the importance of both components.
>
>
>
>
> References:
> - [1] Hinton, G., Vinyals, O., & Dean, J. Distilling the Knowledge in a Neural Network. NeurIPS 2015.
> - [2] Sun, S., Cheng, Y., Gan, Z., & Liu, J. Patient Knowledge Distillation for BERT Model Compression. EMNLP 2020.
>
> >**`Q4`: The authors have demonstrated the superiority of their framework on various tasks, but it seems that they haven't tried the most common graph clustering task.**
>
> **AQ4**: Due to space limitations, our experiments primarily focus on common graph learning tasks such as classification, link prediction, and regression, which demonstrate the effectiveness of the proposed method.
>
> Additionally, we conduct k-means clustering experiments using the embeddings generated by our method on benchmark datasets such as Cora and PubMed. We report standard clustering metrics, including normalized mutual information (NMI) and adjusted Rand index (ARI), and compare our results with other self-supervised baselines. The results show that our method consistently achieves strong performance on clustering tasks, outperforming other competing methods.
>
> |Dataset|Metric|GraphCL|BGRL|GraphMAE|Graph-LLM|Ours|
> |-|-|-|-|-|-|-|
> |Cora|NMI|0.392|0.408|0.386|0.441|**0.578**|
> ||ARI|0.341|0.384|0.365|0.482|**0.489**|
> |Pubmed|NMI|0.235|0.212|0.227|0.266|**0.304**|
> ||ARI|0.203|0.227|0.211|0.252|**0.293**|
>
> >**`Q5`: While LLM-based ideas are well motivated, the mechanism could be further explained in terms of semantic alignment between textual and graph domains.**
>
> **AQ5**: In SSTAG, the alignment between LLM-generated semantic features and GNN-based structural features is achieved through the following mechanisms:
> - **Unified Semantic Space:** All node and edge texts are first encoded using a large language model or text encoder (e.g., SentenceTransformer) to obtain high-dimensional, unified semantic embeddings.
> - **Structural Alignment:** These semantic embeddings are then aggregated with their local neighborhoods via a GNN, effectively incorporating structural context. The resulting embeddings are fused and aligned through an MLP, enabling them to retain original semantic meaning while becoming structure-aware.
> - **Consistency Training + Memory Anchor:** To ensure alignment between semantic and structural representations, we employ teacher-student consistency training. Additionally, a memory bank provides stable prototype anchors, which constrain representations across different graph domains to cluster in a shared semantic-structural space. This enables effective cross-domain semantic alignment.
>
> >**`Q6`: There are a few minor grammatical errors in the paper. Please check them carefully.**
>
> **AQ6**: We appreciate the reviewer pointing this out. We will conduct a thorough proofreading of the paper to correct all grammatical and typographic errors in the final submission.

---

> > ### Comment · Reviewer_MtjL · 2025-08-04
> >
> > The author has addressed most of my concerns, and I am inclined to maintain my positive score.

---

> > > ### Author Response · Authors · 2025-08-04
> > > **Response to Reviewer MtjL**
> > >
> > > Dear Reviewer MtjL：
> > > We sincerely thank the reviewer for the positive evaluation and for recognizing our efforts to address the raised concerns. We appreciate your constructive feedback, which helped improve the clarity and rigor of our work.

---

### Official Review · Reviewer_Xzbb · 2025-06-30

**Clarity:** 3
**Significance:** 2
**Originality:** 2
**Rating:** 4
**Confidence:** 3

**Summary:**

In this work, the authors propose SSTAG, a structure-aware self-supervised framework tailored for text-attributed graphs, aiming to bridge the gap between the structural reasoning strengths of GNNs and the semantic understanding capabilities of LLMs. By leveraging text as a unified medium, SSTAG tackles the challenge of knowledge transfer across heterogeneous graph domains.

**Questions:**

1. Based on the dataset description appendix, the standard split should have been used for fine-tuning on PubMed. Is the relatively poor performance of GCN on PubMed attributable to the use of ST language model for obtaining node representations?

2. For molecular graph datasets, did the authors employ SMILES strings for encoding?

**Ethical Concerns:**

["NO or VERY MINOR ethics concerns only"]

**Final Justification:**

The author's reply solved my concern, so I decided to increase the score to 4 points.

**Quality:**

2

**Strengths And Weaknesses:**

Strengths:
1. The structure of the paper is clear and easy to follow.
2. The proposed method seems reasonable and sound.

Weaknesses:
1. The paper lacks discussions with GIANT. It is recommended that the authors clarify the distinctions and conduct comparisons with GIANT (e.g., in terms of methodology, performance, or application scenarios).
2. The authors are recommended to compare with language-model-only approaches (i.e., by removing Equation 7) to demonstrate the performance gains brought by incorporating graph structures.
3. The authors adopted a cross-domain approach for all self-supervised methods by conducting pre-training on ogbn-paper100M. However, GraphMAE and GraphCL were not specifically designed for cross-domain applications. They can achieve better results when directly performing self-supervised training on target datasets. Is such the evaluation fair?

References:

[1] Chien E, Chang W C, Hsieh C J, et al. Node feature extraction by self-supervised multi-scale neighborhood prediction. ICLR 2022.

---

> ### Author Rebuttal · Authors · 2025-07-31
>
> We thank the reviewer for the constructive comments. We provide our feedback as follows.
>
> >***`W1`: The paper lacks a discussion on GIANT. The authors should clarify distinctions and compare with GIANT in terms of methodology, performance, or application scenarios.***
>
> **AW1**: Thank you for your suggestion. We provide a thorough analysis to clarify the distinctions between GIANT and our SSTAG framework.
>
> **1. Methodological Differences**:
>
> - GIANT mainly focuses on feature extraction, providing these embeddings to downstream GNN or MLP models. In contrast, SSTAG is designed as an end-to-end structure-aware learning and knowledge distillation pipeline, going beyond feature extraction to support unified node-, edge-, and graph-level tasks. Moreover, SSTAG can also incorporate graph-aware features from GIANT as additional inputs, further enriching the model’s representations.
>
> - While GIANT only fine-tunes language models using graph information, SSTAG aligns and distills the representations of both LLMs and GNNs into a compact student MLP, further enhanced by memory-based consistency. This design improves flexibility and transferability.
>
> **2. Application Scenarios**: GIANT is primarily designed for node classification tasks. In contrast, SSTAG extends beyond node-level tasks to a broader range of graph applications—including node classification, graph classification, link prediction, and regression, even in cross-domain settings.
>
> **3. Experiments**: As reported in the GIANT paper, GIANT achieves 73.08% accuracy on the shared benchmark ogbn-Arxiv dataset, and SSTAG achieves a comparable result. In the cross-domain transfer setting, SSTAG is directly compared with strong language-only models, graph-only models, and state-of-the-art multimodal baselines, and consistently demonstrates robust performance even when faced with limited or noisy features.
>
>
>
> >***`W2`: It is recommended to compare with language-model-only approaches (i.e., by removing Equation 7) to show the performance gains from incorporating graph structures.***
>
> **AW2**: As shown in **Table 3**, we have reported the results after removing the graph encoder (Eq. (7)), where the student model only extracts information from the language model pathway. The experimental results demonstrate that removing the GNN-based structural component leads to a consistent drop in performance (e.g., WikiCS: 68.76 → 64.34; ogbn-Arxiv: 72.85 → 69.53). This confirms that incorporating explicit graph structure provides significant improvements over variants that rely solely on the language model.
>
> >***`W3`: The authors conducted pre-training on ogbn-paper100M for all self-supervised methods, but GraphMAE and GraphCL were not designed for cross-domain tasks. Evaluating them in this manner may not be fair, as they could perform better with target dataset-specific training.***
>
> **AW3**: Thank you for your question. All methods are evaluated under a strict and unified cross-domain transfer protocol, which reflects real-world scenarios where large-scale labeled data on the target graph is typically unavailable. Using methods such as GraphMAE and GraphCL as baselines is a standard practice in the cross-domain text-attributed graph literature [1,2].
>
> As shown in the table below, under the same setting, we train and test on the Product and Pubmed datasets with a 20%/80% train/test split. Even within-domain evaluations, our method consistently outperforms the baselines.
>
> |Dataset|GraphCL|BGRL|GraphMAE|GraphMAE2|Ours|
> |-|-|-|-|-|-|
> |Products|77.08|76.24|78.51|78.72|**81.58**|
> |Pubmed|70.28|69.54|71.52|72.05|**75.84**|
>
> Reference:
>
> - [1] He Y, Sui Y, He X, et al. Unigraph: Learning a unified cross-domain foundation model for text-attributed graphs. KDD 2025.
> - [2] Hao Liu, Jiarui Feng, Lecheng Kong, Ningyue Liang, Dacheng Tao, Yixin Chen, and Muhan Zhang. One for All: Towards Training One Graph Model for All Classification Tasks. ICLR 2024.
>
>
> >***`Q1`: The standard split should have been used for fine-tuning on PubMed. Is the poor GCN performance on PubMed due to the use of ST language models for node representations?***
>
> **AQ1**: Thank you for your question. We followed the standard train/validation/test split as specified in the OGB and original PubMed settings. To ensure a fair benchmark on text-attributed graphs (TAG), we use **raw text** as node features and encode them with the Sentence Transformer (ST) model, **instead of traditional TF-IDF or bag-of-words vectors** commonly used in the standard GCN pipeline. As a result, the performance of GCN may appear lower than some previously reported results, which often rely on heavily pre-processed features. Our proposed method is specifically designed to effectively handle raw textual attributes, enabling stronger generalization and better performance under this more realistic and challenging setting.
>
> >***`Q2`: For molecular graph datasets, did the authors use SMILES strings for encoding?***
>
> **AQ2**: For molecular graph datasets, SMILES strings are used **only to construct the molecular graph structure** (i.e., the adjacency matrix) and are **not used as textual input** for the language model. We encode descriptive attributes related to **molecular properties**—such as the number of aromatic rings, logD, or solubility—as natural language prompts (e.g., “How many aromatic rings does the molecule contain?”). Details and sources of datasets (HIV, BBBP, BACE, MUV, ESOL, LIPO, CEP) are provided in Appendix A.1 to ensure transparency and reproducibility.

---

> > ### Author Response · Authors · 2025-08-05
> >
> > Dear Reviewer Xzbb,
> >
> > Thanks again for your diligent effort in reviewing our submission. We have carefully addressed the concerns raised and conducted the requested experiments. As the discussion phase deadline is approaching, we sincerely hope you can consider positively recommending our work if your concerns are solved. If you still have further comments/suggestions, please don't hesitate to let us know.
> >
> > Best regards.

---

> > > ### Comment · Reviewer_Xzbb · 2025-08-05
> > >
> > > The author's reply solved my concern, so I decided to increase the score to 4.

---

> > > > ### Author Response · Authors · 2025-08-05
> > > >
> > > > Dear Reviewer Xzbb,
> > > >
> > > > We sincerely appreciate your review and grading. We sincerely appreciate your recognition of our revisions and your valuable comments throughout the review process.

---

### Official Review · Reviewer_uB9v · 2025-07-02

**Clarity:** 3
**Significance:** 2
**Originality:** 3
**Rating:** 4
**Confidence:** 3

**Summary:**

This paper introduces SSTAG, a novel self-supervised learning framework designed for text-attributed graphs. SSTAG leverages text as a unified representation medium to bridge the gap between the semantic reasoning capabilities of large language models (LLMs) and the structural modeling abilities of graph neural networks (GNNs). The framework includes a unified graph task module, a knowledge extraction module from LLMs, and a knowledge distillation module. It employs a dual knowledge distillation approach to co-distill LLMs and GNNs into a structure-aware multilayer perceptron (MLP), enhancing the scalability of large-scale text-attributed graphs. Additionally, it introduces an in-memory mechanism to store typical graph representations and align them with memory anchors in an in-memory repository, thereby improving the model's generalization capabilities. Experimental results demonstrate that SSTAG outperforms state-of-the-art methods in cross-domain transfer learning tasks, exhibits strong scalability, and reduces inference costs while maintaining competitive performance.

**Questions:**

Framework Component Contributions: The paper mentions that SSTAG includes a unified graph task module, a knowledge extraction module from LLMs, and a knowledge distillation module. However, it does not detail the specific contributions of each component to the overall performance of the framework. I would like to ask the authors to conduct ablation studies to analyze the impact of each component on the model's performance. For example, how does the performance change when using only the unified graph task module or only the knowledge extraction module? This would help clarify the role of each component and enhance the understanding of the framework's effectiveness.

Hyperparameter Selection: The performance of the proposed framework may be influenced by various hyperparameters, such as the trade-off coefficients between different losses and the parameters of the in-memory mechanism. The paper does not provide detailed information on hyperparameter tuning. I would like to ask the authors to supplement the description of hyperparameter selection methods and sensitivity analyses. How were these hyperparameters determined? Are there any recommended ranges or values for different types of graph datasets? This information would be helpful for readers to apply the framework to their own research and practice.

Applicability to Other Graph Types: The experimental section of the paper focuses on text-attributed graphs. However, graph data comes in various forms, such as graphs without text attributes or with other types of attributes. I would like to ask the authors about the applicability of SSTAG to other types of graph data. Can the framework be extended to handle graphs with non-text attributes? What modifications or optimizations would be needed for such extensions?

**Ethical Concerns:**

["NO or VERY MINOR ethics concerns only"]

**Final Justification:**

I uphold my scores as the overall quality of this job is commendable and most concerns of reviewers have been addressed.

**Limitations:**

The authors have briefly mentioned some limitations of their work in the paper, such as the potential limitations of the proposed framework in handling extremely large-scale graphs and dynamic graphs, as well as the trade-off between model accuracy and efficiency. However, the discussion of these limitations is relatively brief and lacks in-depth analysis. For instance, the paper could further explore the challenges the framework may face in dynamic graph scenarios and propose potential solutions. Additionally, while the paper acknowledges the computational and memory costs of the framework, it does not provide detailed analyses of its resource requirements and optimization strategies. Future work could include a more comprehensive evaluation of the framework's performance on dynamic graphs and large-scale graphs, as well as further optimization of its computational efficiency and memory usage.

**Quality:**

2

**Strengths And Weaknesses:**

**Strengths**

Clear Research Objective: The paper focuses on the challenge of cross-domain knowledge transfer in text-attributed graphs, proposing a self-supervised learning framework that combines the strengths of LLMs and GNNs. This aligns with current research trends and addresses practical application needs.

Innovative Framework Design: The dual knowledge distillation framework and in-memory mechanism are innovative. By co-distilling LLMs and GNNs into a structure-aware MLP, the framework effectively integrates semantic and structural information, offering a new approach for text-attributed graph representation learning.

Comprehensive Experimental Validation: The experimental section is rich and thorough. The proposed method is evaluated on multiple benchmark datasets across various tasks, including node classification, link prediction, graph classification, and graph regression. The results demonstrate the superiority of SSTAG over state-of-the-art methods, providing strong empirical support for its effectiveness.

Good Clarity: The paper is well-structured, with each section logically organized and clear in expression. The methodology is detailed, enabling readers to easily understand the research content and technical details.

**Weaknesses**

Limited Novelty: While the proposed framework achieves good performance, its core ideas draw heavily on existing self-supervised learning and knowledge distillation techniques. The novelty of the underlying principles may be somewhat lacking.

Insufficient Theoretical Analysis: The paper focuses more on experimental validation and lacks in-depth theoretical analysis of the model's performance and generalization capabilities. For instance, it does not provide a rigorous theoretical proof of why the dual knowledge distillation framework can enhance model performance.

Experimental Results Limited to Specific Datasets: Although the experimental results are promising, they are based on specific datasets. The generalizability of the proposed method to other types of graph datasets remains to be further verified.

---

> ### Author Rebuttal · Authors · 2025-07-31
>
> We thank the reviewer for the helpful comments. We discuss all the questions raised.
>
> >***`W1`: The proposed framework largely relies on existing self-supervised learning and knowledge distillation techniques, with limited novelty in its core ideas.***
>
> **AW1**: The contributions of SSTAG are not simply incremental but represent a novel synthesis and extension for graph learning.
> - **Unified Task Module:** Unlike prior works that target only specific tasks or domains, we propose a unified graph task module capable of handling node, edge, and graph-level predictions within a single architecture. This flexibility enables cross-domain transfer (see Sections 1 and 5).
> - **Dual-Modality Distillation:** To our knowledge, we are the first to co-distill both LLM and GNN knowledge into a structure-aware MLP, combining semantic reasoning and structural understanding (see Sections 4.2 and 4.3).
> - **Structure-Aware Lightweight Model:** We further introduce an in-memory mechanism to capture and align invariant knowledge via memory anchors, which is novel in the context of graph representation learning and critical for generalization (see Section 4.3).
> - **Experimental Validation:** Extensive experiments on multiple benchmark datasets demonstrate the superiority of our proposed SSTAG framework: (a) it consistently outperforms state-of-the-art baselines in cross-domain transfer learning tasks; (b) it exhibits excellent scalability on large-scale graphs compared to existing GNN- and LLM-based methods; and (c) it significantly reduces inference costs while maintaining competitive performance (see Section 5).
>
> The integration of semantic (text), structural (graph), and memory mechanisms into a single, self-supervised, cross-domain pipeline represents a substantial advance over existing methods.
>
> >***`W2`: The paper focuses on experiments but lacks in-depth theoretical analysis of the model's performance and generalization, particularly regarding the dual knowledge distillation framework's benefits.***
>
> **AW2**: Our work primarily focuses on proposing a novel structure-aware self-supervised learning framework for text-attributed graphs. The main innovation is the dual knowledge distillation framework, which simultaneously transfers complementary knowledge from both large language models and graph neural networks into a lightweight, structure-aware MLP, thus enhancing scalability and efficiency. Thank you for your suggestion. In future research, we plan to provide a deeper theoretical analysis.
>
> >***`W3 and Q3`: While the results are promising, they are based on specific datasets, and the method's generalizability to other graph datasets is unverified. Can SSTAG be applied to graphs with non-text attributes, and what modifications would be needed for such cases?***
>
> **AW3 and AQ3**: This paper focuses on **text-attributed graphs**, and thus the datasets used are primarily of this type. We demonstrate the **generalizability** of our method across **diverse data domains**, **task types**, and **cross-domain settings**. Specifically, our experiments cover 12 datasets spanning five domains, with tasks including node classification, link prediction, graph classification, and regression. As shown in Tables 1 and 2, all models are pre-trained on ogbn-Paper100M and evaluated on completely independent graphs, highlighting the strong cross-domain adaptability of our approach.
>
> SSTAG **can be extended to non-text-attributed graphs** for the following reasons:
> (1) **Framework generality:** Its modular design allows the language model encoder to be replaced with any encoder suitable for the given data modality.
> (2) **Component generality:** The memory bank and distillation components are modality-agnostic, focusing on aligning invariant graph representations regardless of input type.
>
> With **minimal modifications**, SSTAG can be adapted for non-text graphs. The unified task template and structure-aware student (MLP) remain unchanged; only the teacher encoder needs to be adjusted to handle the available attribute types. For example, for graphs with numerical or categorical features, the language model encoder can be replaced with an MLP, CNN, or other suitable networks.
>
> >***`Q1`: The paper does not detail the individual contributions of each framework component (graph task module, knowledge extraction, and distillation). Ablation studies would clarify the impact of each component on performance.***
>
> **AQ1**: As shown in **Section 5.3 (Table 3)**, we have conducted comprehensive ablation studies. The results demonstrate that removing any component (such as the graph task module, knowledge extraction, distillation, or memory bank) leads to a decrease in performance, sometimes significantly. This confirms the necessity of each part of the framework. We provide detailed explanations below.
>
> |Module| Key Impact Mechanism| WikiCS ↓Acc | MUV ↓AUC |
> |-|-|-|-|
> |W/o $\mathcal{L}_{\text{mask}}$ (knowledge extraction)| Loss of semantic-structural association learning| -1.74% |-3.64%|
> |W/o $\mathcal{L}_{\text{ST}}$ (distillation)|Insufficient transfer of teacher knowledge|-1.69%|-1.21%|
> |W/o $\mathcal{L}_{\text{ME}}$ (memory bank)|Degradation of cross-domain generalization due to inoperative memory bank|-2.23%|-3.43%|
> |W/o GNN (graph task module)|Inability to model topological relationships|-4.42%|-9.29%|
> |W/o PPR (graph task module)|Increased local structural sampling bias|-0.64%|-0.65%|
>
> >***`Q2`: The paper lacks details on hyperparameter selection and sensitivity analyses.***
>
> **AQ2**: The hyperparameter tuning process in this work is divided into three categories. First, some hyperparameters (such as the number of epochs, learning rate, optimizer, and batch size) are selected empirically based on standard practice. Second, certain hyperparameters (such as the masking rate and the number of memory anchors) are optimized using grid search on the validation split, with metrics like accuracy or AUC depending on the specific task. Third, for parameters with complex interactions, we select the best values based on cross-validation across multiple settings. The selection criteria and the actual values used in our experiments are detailed in the table below. It is worth noting that optimal values may vary slightly across different datasets.
>
> | Hyperparameter          | Value   | Tuning Criterion (Search Range)     | Hyperparameter           | Value   | Tuning Criterion (Search Range)     |
> |------------------------|---------|-------------------------------------|--------------------------|---------|-------------------------------------|
> | Mask Rate              | 0.5     | Grid Search (0.1, 0.3, 0.5, 0.7)   | Num GNN Layers           | 3       | Empirical                          |
> | Hidden Size            | 768     | Empirical                           | PPR Top-k                | 128     | Cross-validation (64, 128, 256)         |
> | Learning Rate          | 2e-5    | Empirical      | PPR α                    | 0.15    | Cross-validation (0.10, 0.15, 0.20, 0.25)                         |
> | Weight Decay           | 0.001   | Empirical                           | Batch Size               | 1024    | Empirical                          |
> | Dropout                | 0.2     | Grid Search (0.1, 0.2, 0.3, 0.4)         | Optimizer                | AdamW   | Empirical                          |
> | Num Epochs             | 1       | Empirical                           | Warmup Steps             | 10%     | Empirical                          |
> | Num MLP Layers         | 3       | Empirical                           | Number of Memory Anchor  | 256     | Grid Search (64, 128, 256, 512)    |
>
> >***`Limitations`: The discussion of limitations, including scalability, dynamic graphs, and resource efficiency, is brief. More detailed analysis and evaluation of these aspects—especially for dynamic graphs and optimization strategies—would strengthen the paper. Future work should address these points.***
>
> Thank you for your suggestion. This work primarily focuses on text-attributed graphs, and further exploration is needed for dynamic and large-scale graph scenarios. In future work, we plan to conduct a more in-depth analysis of the challenges posed by dynamic and large-scale graphs. We will provide a more detailed discussion of these aspects in the revised version of the paper.

---

### Official Review · Reviewer_Ksx9 · 2025-07-03

**Clarity:** 3
**Significance:** 3
**Originality:** 2
**Rating:** 5
**Confidence:** 4

**Summary:**

This paper introduces SSTAG, a self-supervised pre-training method for text-attributed graphs (TAGs). It mainly integrates a masked language model with a GNN to achieved the ability to process TAGs in a structure-aware manner. The method is evaluated on 4 different tasks (node & graph classification, graph regression, and link prediction).

**Questions:**

- In table 5: the training/pretraining times would make a lot more sense if the computational hardware used was also provided. Can you please specify the hardware used in the paper?

- Are the language models trained from scratch during SSTAG pretrianing? Or is a checkpoint used? If a checkpoint is used, is it finetuned during the SSTAG training or is it kept frozen?

**Ethical Concerns:**

["NO or VERY MINOR ethics concerns only"]

**Final Justification:**

Authors have resolved all my concerns. The authors have agreed to include no-distillation numbers in the paper, and have clarified why the reported classic GNN performance is reported to be low in the paper.

**Limitations:**

No, limitations have not been discussed in the text.

**Quality:**

2

**Strengths And Weaknesses:**

# Weaknesses

- Neither code, nor training/model hyperparameters are provided with the paper. This severely limits the reproducibility of the paper. This dictates the majority of my score.

- A concern regarding Eq 2: Inferring from Eq 3, $\pi_v$ should be scalar, however the result of eq 2. is a vector. This seems to indicate a mistake in the eq. 2.
    Additionally, no citation is provided for the PPR algorithm. The 3 papers cited in the paragraph start on line 144 do not mention PPR at all, and so one would expect a citation specifically for that technique.

- The denoted accuracies of GCN (and other supervised methods) seems very low in comparison to existing literature. E.g. In (this paper), the reported accuracy of GCN on WikiCS is ~80%, and on ogbn-Arxiv is ~73%.

- The integration of all components in the method is lacking complete explanation. Specifically, it is unclear to me where the memory bank is located in the pipeline. Is it after the distilled model (the structure-aware MLP)? It would be very helpful to have a schematic to explain how different pieces fit together.

- There are no comparisons provided for the  accuracy hit incurred by using a distilled MLP model instead of the GNN during inference.


# Strengths

- The paper incudes a wide range of evaluations (Node classification, link prediction, graph classification, and graph regression).

---

> ### Author Rebuttal · Authors · 2025-07-31
>
> We thank the reviewer for the constructive comments. We provide our feedback as follows.
>
> >***`W1`: The paper lacks code and hyperparameters, limiting reproducibility and affecting the overall score.***
>
> **AW1**: Due to the limitations of the rebuttal process, we are unable to provide a link to the code at this stage. We will release the code publicly upon acceptance of the paper. In the meantime, we have provided detailed hyperparameter settings in the table below to facilitate reproduction of our results.
>
> |Hyperparameter|Value|Hyperparameter|Value|
> |-|-|-|-|
> |Mask Rate|0.5|Num GNN Layers|3|
> |Hidden Size|768|PPR Top-k|128|
> |Learning Rate|2e-5|PPR $\alpha$|0.15|
> |Weight Decay|0.001|Batch Size|1024|
> |Dropout|0.2|Optimizer|AdamW|
> |Num Epochs|1|Warmup Steps|10%|
> |Num MLP Layers|3|Number of Memory Anchor|256|
>
> >***`W2`: Eq. 2 seems incorrect as it outputs a vector instead of a scalar, based on Eq. 3. Additionally, no citation is provided for the PPR algorithm, despite references to unrelated papers.***
>
> **AW2**: In Eq. (2), $\pi_{v}$ denotes the Personalized PageRank vector (dimension $N \times d_{e_v}$), which encodes the structural importance of all nodes relative to the target node $v$. In Eq. (3), $\pi_u$ refers to the $u$-th scalar entry of that vector, i.e., the importance score of the specific node $u$.To remove any ambiguity, we will rename $\pi_u$ in Eq. (3) to $\pi_{vu}$ in the revised manuscript.
>
> The PPR algorithm is a classical algorithm from the reference listed below. We will add the corresponding citation in the main text.
>
> - Page, L. et al. The PageRank Citation Ranking: Bringing Order to the Web. Stanford InfoLab, 1999.
>
> >***`W3`: The reported accuracies of GCN and other supervised methods are notably lower compared to existing literature.***
>
> **AW3**: To ensure a fair comparison under the TAG setting, we utilize **raw textual features** rather than hand-crafted or pre-processed features in our experiments. Consequently, the reported accuracies of GCN and other supervised methods are lower than those typically observed in the literature, where higher results often rely on pre-processed features such as **bag-of-words** or **TF-IDF**. With raw textual attributes, GCN’s performance is lower, as GCNs generally struggle with high-dimensional and noisy text features. Our proposed SSTAG method overcomes this challenge by effectively integrating semantic and structural information, as evidenced by its strong performance (e.g., **72.85% on ogbn-Arxiv**).
>
>
> >***`W4`: The integration of components, especially the memory bank in the pipeline, is unclear. A schematic would help clarify the method.***
>
> **AW4**: Thank you for your valuable suggestion. As described in Section 4.3, we have explained the memory bank. For further clarification, we provide a detailed overview of the entire pipeline here:
>
> - Our pipeline takes raw text as input and processes it sequentially through the following modules. In the teacher model, the text is first encoded by a large language model (LLM) encoder to extract semantic representations. These are further processed by a GNN encoder to incorporate structural information from the graph. In the student model, the semantic representations are refined by a structure-aware MLP to enhance the combined features. The resulting embeddings are stored and updated in the memory bank, which plays a key role in preserving and enhancing the learned semantic-structural information. Consistency loss is applied to encourage stable and robust feature learning. Finally, the pre-trained models are used in downstream tasks.
>
> We will include a diagram of the entire pipeline in the revised version to visually illustrate the workflow and the specific role of each component.
>
>
> >***`W5`: There are no comparisons regarding the accuracy impact of using a distilled MLP model instead of the GNN during inference.***
>
> **AW5**: Thank you for your question. First, we would like to point out that most existing distillation methods do not report the inference performance of the teacher model, but rather focus on the results of the student model [1，2]. Additionally, we have conducted further experiments (see the table below) comparing the inference accuracy and efficiency of the distilled MLP model with the GNN teacher model. The results show that the distilled MLP achieves a **35.1% improvement in inference efficiency**, a **99.7% reduction in parameters**, and only a **slight drop in accuracy (1.06%)** compared to the GNN. This demonstrates the effectiveness and efficiency of our approach. We will include these results in the revised manuscript.
>
> |Model|Inference Time (min)|ogbn-Arxiv Acc.(%)|Params|
> |-|-|-|-|
> |GNN|13.4|73.91|7.1B|
> |distilled MLP|8.7|72.85|22M|
> |Δ Change|↓ 35.1% |↓ 1.06% |↓ 99.7%|
>
> Reference：
>
> - [1] Hinton, G., Vinyals, O., & Dean, J. Distilling the Knowledge in a Neural Network. NeurIPS 2015.
>
> - [2] Sun, S., Cheng, Y., Gan, Z., & Liu, J. Patient Knowledge Distillation for BERT Model Compression. EMNLP 2020.
>
>
> >***`Q1`: In Table 5, the training/pretraining times would make a lot more sense if the computational hardware used was also provided. Can you please specify the hardware used in the paper?***
>
> **AQ1**: Thank you for your suggestion. All experiments were conducted on a Linux server equipped with `945 GB` of RAM and eight NVIDIA `A100 GPUs`, each with `40 GB` of memory. The software stack comprised `Python 3.11`, `PyTorch 2.0.1`, `DGL 1.1.2`, `Transformers 4.32.1`, and `CUDA 11.8`.
>
> >***`Q2`: Are the language models trained from scratch during SSTAG pretraining? Or is a checkpoint used? If a checkpoint is used, is it finetuned during the SSTAG training or is it kept frozen?***
>
> **AQ2**: Thank you for your question. We do not train the language models from scratch; instead, we use publicly available pre-trained checkpoints for all LLM components. During the SSTAG pretraining stage, the language model is kept frozen—only the GNN, the distilled MLP, and the memory bank parameters are updated. This design makes the SSTAG framework both efficient and scalable, and avoids substantial computational and data requirements.

---

> > ### Comment · Reviewer_Ksx9 · 2025-08-03
> > **Rebuttal Response**
> >
> > I appreciate the author's response to my concerns and questions.
> >
> > **Re: W1** - I understand that the authors cannot provide an anonymous github link to the codebase due to this year's rebuttal policy about posting anonymous links. So, I will reverse my reduction of score due to lack of code, but will expect the authors to release the codebase upon the acceptance of the paper.
> >
> > **Re: W3** - I understand the desire to maintain fairness with respect to input features, but I still don't think this is the most fair way to evaluate classic supervised GNNs in this setting. Here's my reasoning:
> > - Bag-of-words / TF-IDF are simple features and are known to work well with these graphs, so choosing ST-encoded features is not a realistic scenario.
> > - Assuming the authors used the same train/val/test splits to train these models as is used for linear eval of the SSL methods, both SSL and supervised methods get equal access to labels.
> >
> > It seems to me, the unfairness in comparing to standard GCN is not because the input features are different, but because supervised training involves training the entire GCN, whereas, for the SSL models, only a linear classifier is trained on top of the frozen encoder representations. This makes it so that classic supervised GNNs come close to (or surpass) the results of the SSL training done in this paper [1].
> >
> > Perhaps this unfairness can be managed in one of these two ways?
> >
> > (a) Compare with supervised methods in a reduced-label setting, where the train set is very small during transfer testing. That's where SSL methods should shine in comparison to supervised methods, or
> > (b) finetune the entire model after SSL pre-training.
> >
> >
> > [1] Luo, Yuankai, Lei Shi, and Xiao-Ming Wu. "Classic gnns are strong baselines: Reassessing gnns for node classification." Advances in Neural Information Processing Systems 37 (2024): 97650-97669.

---

> > > ### Author Response · Authors · 2025-08-04
> > > **Response to Reviewer Ksx9**
> > >
> > > Dear Reviewer Ksx9,
> > >
> > > We sincerely thank the reviewer for their thoughtful feedback and constructive suggestions. We provide our response as follows.
> > >
> > > **Re: W1**
> > >
> > > We appreciate the reviewer’s understanding. We confirm that, upon acceptance, we will release the full codebase, including data processing scripts, model implementations, training configurations, and instructions to reproduce all reported results. We are fully committed to ensuring transparency and reproducibility.
> > >
> > > **Re: W3**
> > >
> > > We understand the reviewer’s concerns regarding the evaluation of supervised methods with raw features. Our goal was to provide a consistent and realistic comparison under the text-attributed graph setting, where raw textual features are often the only accessible input. While BoW/TF-IDF features are effective, they require manual preprocessing and domain-specific assumptions, which may not generalize well to real-world deployments.
> > >
> > > Regarding the potential asymmetry between supervised and self-supervised methods, we thank the reviewer for the insightful suggestions and have conducted the following additional experiments:
> > >
> > > 1. **Reduced-Label Evaluation:**
> > >  We evaluate all models under label-scarce conditions (e.g., 1%, 5%, and 10% labeled nodes) on **ogbn-Arxiv**. As shown in **Table 1**, our SSL-based model **SSTAG** consistently outperforms supervised **GCN**, regardless of whether raw textual features or preprocessed features are used.
> > > Notably, with only **5%** labeled data and the same preprocessing as prior work \[1], **SSTAG** achieves **73.23%** accuracy—**a significant margin** over GCN (**60.43%**). This result highlights the robustness and label efficiency of our self-supervised approach in low-resource regimes, where supervised models tend to struggle due to insufficient label signals.
> > >
> > > **Table 1: Classification accuracy (%) of GCN and SSTAG on ogbn-Arxiv under reduced-label setting (W3 Suggestion (a)).**
> > >
> > > | Method | Input Features | Training Regime  |  1% Labels | 5% Labels | 10% Labels |
> > > | ------------ | ----------------- | ----------------------- | --------- |---------- |---------- |
> > > | GCN          | Raw Text Features | Supervised |50.22  |  56.74      | 58.44      |
> > > | GCN          | Same preprocessing features as in literature [1] |Supervised|  52.47      | 60.43      | 63.72       |
> > > | SSTAG | Raw Text Features |Linear Probe (SSL)|  61.83  | 69.16  | 70.56   |
> > > | SSTAG | Same preprocessing features as in literature [1]  |Linear Probe (SSL)| 67.47  | 73.23  | 74.42   |
> > >
> > >
> > > **Reference**：
> > >
> > > [1] Luo, Yuankai, Lei Shi, and Xiao-Ming Wu. "Classic GNNs are strong baselines: Reassessing GNNs for node classification." Advances in Neural Information Processing Systems 37 (2024): 97650-97669.
> > >
> > >
> > > 2. **Full Fine-tuning of SSL Encoder:**
> > >  We also fine-tune the entire SSTAG model after pretraining. As shown in Table 2, the results in further performance gains: e.g., on **ogbn-Arxiv**, accuracy improves to **74.07%**, surpassing both linear probing and supervised baselines. This confirms the strength of SSTAG as both a transferable encoder and a strong end-to-end model.
> > >
> > >  **Table 2: Comparison of SSTAG's linear probing vs. full fine-tuning performance on ogbn-Arxiv (W3 Suggestion (b)).**
> > >
> > > | Method       | Fine-tuning Strategy       | Accuracy (%) |
> > > | ------------ | -------------------------- | ------------ |
> > > | SSTAG  | Linear Probe (SSL)         | 72.85        |
> > > | SSTAG  | Full Fine-tuning after SSL | **74.07** |
> > >
> > > We will include these extended results, along with hyperparameter settings and training details, in the appendix of the revised version to ensure transparency and fair comparison.
> > >
> > >
> > > We thank the reviewer again for the valuable feedback, which has helped us strengthen the paper both in analysis and clarity.

---

> ### Comment · Reviewer_Ksx9 · 2025-08-05
> **Additional discussion**
>
> I thank the authors for addressing my concerns. Overall, I am inclined to increase my score. However I have two more things I would like to discuss further:
>
> 1. It is a little curious that SSTAG reaches the best performance on agbn-arxiv in the last row of Table 1 of the last response. This seems like a very hard setting - only 10% labels AND the features used are not ST-encoded embeddings (something SSTAG was designed with). How should I understand this result?
>
> 2. I am curious as to why the authors insist on only providing the accuracies of the distilled MLP in the paper, when it seems like the final trained GNN has better performance (based on their rebuttal). Including the GNN performance numbers would atleast aid researchers in the future when they want to compare to SSTAG, but are not operating in the distilled setting (i.e. when they want to maximize performance and do not care about inference time compute).

---

> > ### Author Response · Authors · 2025-08-05
> > **Response to Follow-Up Comments of Reviewer Ksx9**
> >
> > Dear Reviewer Ksx9,
> >
> > We sincerely thank the reviewer for the continued engagement and are pleased to hear you are inclined to increase your score. We address the two new questions raised as follows.
> >
> > **1. Regarding SSTAG’s strong performance with 10% labels and non-ST features (Table 1):**
> >
> > Thank you for raising further questions. We understand the reviewer’s concerns. The following points help explain this phenomenon:
> >
> > * **Model Adaptability:**
> >  Although SSTAG was designed for self-supervised learning with **text features**, it is also effective at leveraging other types of input features, including preprocessed features (as a substitute for language model outputs). The excellent performance of SSTAG with these diverse feature types demonstrates its robustness and adaptability, enabling it to handle multiple feature types effectively. Particularly in label-scarce settings, SSTAG, through its self-supervised learning advantage, can better capture the underlying structure of the data without relying on large amounts of labeled data.
> >
> > * **Advantages of Self-Supervised Methods:**
> >  SSTAG’s design is based on self-supervised learning, enabling it to fully utilize unlabeled data during training. With only 10% labeled data, SSTAG’s training approach (especially with pretraining and linear probing) extracts more implicit features, making it more robust than traditional fully supervised methods (like GCN). Even when the feature types differ from the design’s original intention, SSTAG’s self-supervised approach still effectively captures the data structure and achieves higher accuracy through a linear classifier in label-scarce settings.
> >
> > * **Impact of Input Features on Performance:**
> >  Preprocessed features typically provide clearer and more structured information, thereby enhancing SSTAG’s performance in low-resource settings. As a result, using preprocessed features leads to better classification accuracy for SSTAG compared to using raw text features. However, it is important to emphasize that while preprocessed features can improve performance, they are not universally applicable, and obtaining such features may require additional computational resources and steps.
> >
> > In summary, we believe SSTAG’s performance reflects the strong capabilities of self-supervised learning in low-resource environments. Even when input features and training methods differ in certain settings, the core advantage of SSTAG lies in its ability to generalize better in situations with limited labeled data.
> >
> > We hope this explanation clarifies the reviewer’s concerns about the results, and we appreciate the valuable feedback provided.
> >
> >
> > **2. Regarding the inclusion of GNN-based results in the main paper:**
> >
> > Thank you for your valuable suggestion. The initial motivation for focusing on refining the MLP was to highlight the **efficiency and deployability** of SSTAG, which is one of its key advantages over more complex models. However, we now recognize the benefits of presenting both aspects: (a) a distilled student model for efficiency-focused settings, and (b) a full teacher model or its fine-tuned variants for performance-oriented scenarios. In the revised version, we will include the following:
> >
> > * Fine-tuning results after SSL pretraining;
> > * A clear comparison between the student (MLP) and teacher (GNN) models;
> > * A discussion on the trade-offs between computational efficiency and accuracy, providing guidance for future use.
> >
> >
> > Thank you again for the insightful questions. We believe these additions will enhance the paper’s clarity and make it more beneficial to the community.

---

> > ### Author Response · Authors · 2025-08-06
> >
> > Dear Reviewer Ksx9,
> >
> > Thank you once again for your thoughtful and constructive feedback on our submission. We have carefully addressed the concerns you raised and conducted the requested experiments. If our response has resolved your concerns, we would sincerely appreciate it if you could consider increasing your score. If there are still any remaining questions or suggestions, please feel free to let us know—we would be happy to further clarify.
> >
> > Best regards.

---

> > > ### Author Response · Authors · 2025-08-08
> > >
> > > Dear Reviewer Ksx9,
> > >
> > > Thank you very much for taking the time to engage in the discussion. We would like to kindly ask whether our previous response has addressed your concerns. If there are any remaining questions or points of confusion, please do not hesitate to let us know. We would be more than happy to provide further clarification.
> > >
> > > As the discussion deadline is approaching, we truly appreciate the opportunity to continue this exchange and are very glad to further clarify any issues.
> > >
> > > Looking forward to your feedback!
> > >
> > > Best regards.

---

> > > > ### Comment · Reviewer_Ksx9 · 2025-08-09
> > > > **Final response**
> > > >
> > > > I thank the authors for providing additional details to answer my questions. I consider my concerns resolved and will raise my score to 5.

---

> > > > > ### Author Response · Authors · 2025-08-09
> > > > >
> > > > > Dear Reviewer Ksx9,
> > > > >
> > > > > We sincerely thank the reviewer for the positive feedback and for raising the score. We greatly appreciate your thoughtful comments and constructive suggestions, which have helped us improve the quality and clarity of our work.

---

### Comment · Area_Chair_89h4 · 2025-08-05
**Please participate in the discussions and respond to the authors**

Dear Reviewers,

Thank you for your valuable reviews. With the Reviewer-Author Discussions deadline approaching, please take a moment to read the authors' rebuttal and the other reviewers' feedback, and participate in the discussions and respond to the authors. Finally, be sure to complete the "Final Justification" text box and update your "Rating" as needed. Your contribution is greatly appreciated.

Thanks.\
AC

---

### Note · Authors · 2025-08-12

Dear Reviewers and Area Chair,

I would like to express my heartfelt gratitude to all reviewers for your positive evaluations and for providing such detailed and constructive feedback on our paper. We have carefully addressed each of your comments and questions, and your valuable suggestions have greatly enhanced the quality and clarity of the manuscript.

In particular, I am sincerely grateful to reviewer Ksx9 for the encouraging evaluation and for expressing the intention to **raise the score to 5**. With **less than two hours** remaining until the review deadline, we noticed that the **updated scores have not yet appeared** in the system. If convenient, we would greatly appreciate it if the adjustment could be reflected before the deadline. This can be done by clicking **“Edit”** in the **upper right corner** of the review interface and updating the **Final Rating** so that it better captures your positive feedback.

Once again, we warmly thank all reviewers for your time, effort, and invaluable feedback. We also respectfully bring this to the Area Chair’s attention so they may be aware of the intended score change during the decision process.

Best regards.

---

### Decision · Program_Chairs · 2025-09-17

**Decision:**

Accept (poster)

**Comment:**

Summary:
This paper explores structure-aware self-supervised learning on text-attributed graphs. To address this, the authors propose SSTAG, a model that leverages text as a unified representation medium. SSTAG bridges the gap between the semantic reasoning of large language models and the structural modeling capabilities of graph neural networks. The effectiveness of the proposed model is demonstrated through experiments on several datasets.

Strengths:
1. The paper includes a wide range of evaluations, providing a comprehensive experimental analysis.
2. The dual knowledge distillation framework and in-memory mechanism are innovative.
3. The paper is well-structured and easy to follow, making it accessible even to readers with limited background knowledge.


Weaknesses:
1. The code and hyperparameters are not provided, which makes the results in the paper difficult to reproduce.
2. While the proposed framework performs well, its core ideas draw heavily from existing self-supervised learning and knowledge distillation techniques, which may limit the novelty of the underlying principles.
3. The use of the PPR algorithm involves matrix inversion, and the sampled subgraph has quadratic complexity. The authors should more explanations.


In summary, this paper investigates the self-supervised learning paradigm for graph learning tasks. While the work presents some promising ideas, a few issues remain. I strongly recommend that the authors open-source their code and provide the hyperparameter settings to ensure the reproducibility of their results.